# PROOF: Perturbation-Robust Noise Finetune via Optimal Transport Information Bottleneck for Highly-Correlated Asset Generation

## Abstract

The diffusion model has provided a strong tool for implementing text-to-image (T2I) and image-to-image (I2I) generation. Recently, topology and texture control have been popular explorations. Explicit methods consider high-fidelity controllable editing based on external signals or diffusion feature manipulations. The implicit method naively conducts noise interpolation in manifold space. However, they suffer from low robustness of topology and texture under noise perturbations. In this paper, we first propose a plug-and-play **P**erturbation-**RO**bust n**O**ise **F**inetune (*PROOF*) module employed by Stable Diffusion to realize a trade-off between content preservation and controllable diversity for highly correlated asset generation. Information bottleneck (IB) and optimal transport (OT) are capable of producing high-fidelity image variations considering topology and texture alignments, respectively. We derive the closed-form solution of the optimal interpolation weight based on optimal-transported information bottleneck (OTIB), and design the corresponding architecture to fine-tune seed noise or inverse noise with around only 14K trainable parameters and 10 minutes of training. Comprehensive experiments and ablation studies demonstrate that PROOF provides a powerful unified latent manipulation module to efficiently fine-tune the 2D/3D assets with text or image guidance, based on multiple base model architectures.

## 1 Introduction

Controllable T2I and I2I are challenging and meaningful tasks for asset creation. Previous diffusion control models try to implement structure or appearance-aligned generation explicitly, mainly by feature-level modulation Lin et al. (2024); Mo et al. (2024); Epstein et al. (2023), adapter injection Mou et al. (2024); Zhao et al. (2023); Ye et al. (2023), and model fine-tuning based on external structure or appearance signals Zhang et al. (2023a); Gal et al. (2023); Ruiz et al. (2023; 2024). Explicit methods are dependent on cumbersome user control guidance, which hinders topological diversity and appearance robustness as well. On the contrary, we pay attention to the implicit noise-level manipulation on the inherent latent space, where we conduct a trade-off of diversity, structure, and appearance simultaneously.

Recently, test-time noise searching Ma et al. (2025); Zhou et al. (2025) has proved that golden noise plays an important role in diffusion performance for semantic alignment. Other latent manipulation methods, e.g., UnCLIP Ramesh et al. (2022), Kwon et al. (2023), also focus on generating semantic-aligned variants. These works have a fundamental task distinction compared with PROOF. We assume the noise latent has been semantic-aligned, and conduct content-aligned variants with robust structures and textures preservation. We briefly introduce our motivation as follows.

Gaussian noise inherently encodes contextual information. It is supposed to adaptively inject diverse information into the source content while adversarially preserving the original content distribution. This fidelity-diversity trade-off needs to learn a pixel-wise minimal sufficient representation of the noise latent. Inspired by information bottleneck Tishby & Zaslavsky (2015); Schulz et al. (2020), we compress the content latent for topology alignment in an implicit view of the mutual information.

Furthermore, spatial attention is important to improve the contextual perception and appearance robustness. Noise features are distributed randomly without obviously recognizable patterns. Therefore, it is supposed to distribute attention in a coordinated manner to eliminate excessive local atten-

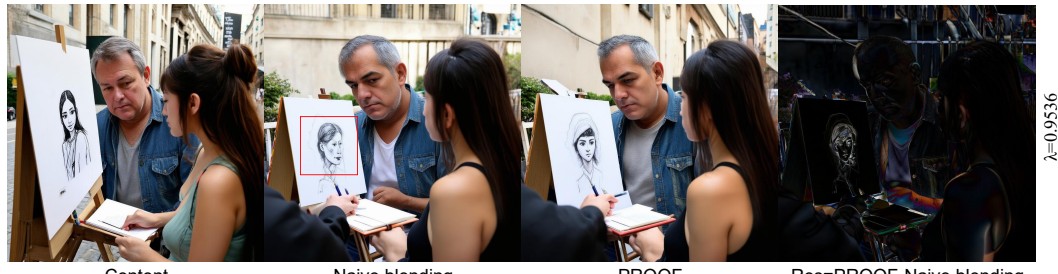

Figure 1: Content-diversity tradeoff: given a noise latent of a content, naive noise blending with interpolation weight $\lambda$ generates uncontrollable topology and appearance. PROOF finetunes noise latent where adaptively injecting the perturbation based on the optimal transported information bottleneck. The structure and appearance statistics from the content are preserved well, with concurrently controllable diversity. **Res** means the optimized area of PROOF compared with naive blending.

tion. However, traditional QKV attention uses Softmax, which lacks this global attention distribution ability. Inspired by Sinkhorn optimal transport Cuturi (2013); Kim et al. (2024), we apply the doubly stochastic activation constraint to better model the global feature relationships in noise space. This optimally transported attention exhibits significant appearance fidelity. More remarkably, we derive the closed-form solution of the Sinkhorn-regularized IB interpolation weight, which is the theoretical foundation of the PROOF architecture. More details are represented in Sec. 4.3.

As shown in Figure 1, the mainstream implicit approach, i.e., naive noise interpolation with a per-pixel constant weight $\lambda$ for original noise and $(1-\lambda)$ for another noise perturbation, fails to preserve the structure and appearance statistics of the original content. In our task definition, the assets for content and naive blending are not highly correlated due to substantial inconsistency of structure and appearance. In contrast, our PROOF adaptively blends pixel-wise perturbations via activation optimization in noise space, based on the proposed Optimal-Transported Information Bottleneck module, thereby facilitating precise asset variations. Our paper presents several significant contributions, mainly including three folds:

1. We first explore the structure and appearance-aligned 2D/3D asset generation by means of perturbation-robust noise representation learning rather than other explicit control manners, such as attention matrices, intermediate activations, or external control signals. Remarkably, test-time *PROOF* demands merely brief training while maintaining full disentanglement from the diffusion model's forward and denoising process.

2. We present an efficient and effective Optimal-Transported Information Bottleneck module that provides a trade-off between content preservation and mode variety. IB prevents the learning from mode collapse, and OT promotes higher faithfulness of textures. Moreover, we derive the closed-form solution of the Sinkhorn-regularized IB interpolation weight. This mathematical derivation is aligned with the information flow of OTIB, which provides a solid theoretical foundation for OTIB.

3. Our proposed PROOF is capable of being adaptive for multiple asset creation tasks, base architectures, and model checkpoints. Compared with state-of-the-art structure and appearance-aligned approaches, comprehensive experimental analyses demonstrate that PROOF is the first perturbation-robust plug-and-play implicit controller for pre-trained T2I models. Furthermore, PROOF is superior to other diversity-inducing methods, such as entropy regularization and contrastive objective.

## 2 RELATED WORK

We briefly introduce diffusion control methods, diffusion seed manipulation, and information compression works in this section.

**Diffusion control.** On one hand, pre-trained T2I foundational models Rombach et al. (2022) are potentially able to generate diverse images taking advantage of the random noise initialization. On the other hand, uncertainty from the Gaussian noises makes it hard to synthesize credible images with a certain topology or texture. To address this matter, previous diffusion control methods compose different adapters independently Mou et al. (2024); Zhao et al. (2023), or conduct adaptively feature modulations Zhang et al. (2023a); Lin et al. (2024), and model finetune Gal et al. (2023); Ruiz et al. (2023) to facilitate alignment of internal diffusion knowledge and external control signals.

*Topology alignment* SD-based methods have demonstrated strong generalization capabilities and composability while maintaining high creation quality Li et al. (2023); Zhao et al. (2023); Yang et al. (2023); Avrahami et al. (2023b); Zheng et al. (2023); Wang et al. (2024); Zhou et al. (2024). External control signals include Canny edge, depth map, human pose, line drawing, HED edge drawing, normal map, segmentation mask (used in Zhang et al. (2023a); Zhao et al. (2023)), as well as 3d mesh, point cloud, sketch (used in Lin et al. (2024)), etc. FreeControl Mo et al. (2024) manipulates the specific-class linear semantic subspace to employ structural guidance. Semantic signal usually possesses higher freedom than low-level vision signals. Note that our PROOF does not depend on any external structure control signal.

*Texture alignment* methods try to realize I2I by image prior embedding or few-shot weight adaptation. General I2I methods extract global semantic embedding from the referenced images Zhao et al. (2023); Ye et al. (2023); Mou et al. (2024). Personalized model concerning specific concept needs pretrained T2I diffusion finetuning based on a small set of image samples Ruiz et al. (2023); Gal et al. (2023); Avrahami et al. (2023a); Po et al. (2024); Ruiz et al. (2024). FreeControl Mo et al. (2024) uses intermediate activations as the appearance representation, similar to DSG Epstein et al. (2023). However, our PROOF achieves superior appearance alignment performance without personalized concept data or model fine-tuning.

**Diffusion seed.** Previous diffusion control methods only treat Gaussian noise as a flexible random generation seed Zhang et al. (2023a); Zhao et al. (2023); Ye et al. (2023); Zheng et al. (2023); Wang et al. (2024); Zhou et al. (2024); Ruiz et al. (2023); Gal et al. (2023); Avrahami et al. (2023a); Po et al. (2024); Ruiz et al. (2024). They constrain the pre-trained diffusion model using external structure or textural data. Nevertheless, some diffusion inversion works Yang et al. (2025); Song et al. (2020); Mokady et al. (2023) show high-fidelity image reconstruction and editing. Seed searching Ma et al. (2025) is beyond the denoising steps for high-quality image generation. These methods establish the critical role of noise representation, which is demonstrated by Figure 1 as well. Therefore, we explore the implicit structure and appearance alignment based on noise in this paper.

**Information bottleneck.** Information bottleneck (IB) Tishby & Zaslavsky (2015) plays a representation trade-off between information compression and information preservation for neural learning tasks. Furthermore, VIB Alemi et al. (2017) leverages variational inference to facilitate the IB neural compression. IBA Schulz et al. (2020); Gao et al. (2021) polishes the attribution information based on KL divergence Csiszár (1975) to effectively disentangle relative and irrelative information concerning the classification task. We will introduce our information bottleneck in Section 3, 4.

## 3 PRELIMINARIES

### 3.1 PROBLEM SETTING

Given source noise $N_{Orig}$ and injected noise $N_{Div}$ are from a consistent distribution $\mathcal{N}(\mu_G, \sigma_G^2)$, where $\mu_G$ and $\sigma_G$ represent the means and standard deviations. Then, the modulated manifold of 2D/3D asset can be formulated as follows Schulz et al. (2020):

$$N_{Out} = \lambda N_{Orig} + (1 - \lambda)N_{Div}, \tag{1}$$

where $\lambda$ is the blending weight as the hyperparameter or learned prior, $N_{Div}$ is the noise perturbation. Given $N_{Out}$ as $z_t$, the latent diffusion model Rombach et al. (2022) conducts a denoising process on the compressed latent from the Gaussian noise distribution. The denoised manifold of the pre-trained diffusion model is calculated as follows:

$$\tilde{z}_0 = \frac{z_t}{\sqrt{\bar{\alpha}_t}} - \frac{\sqrt{1 - \bar{\alpha}_t}\epsilon_\theta(z_t, c, t)}{\sqrt{\bar{\alpha}_t}}. \tag{2}$$

where $\epsilon_\theta$ is the denoising propagation parameter, $t$ is the diffusion timestep, $c$ means prompt, $\alpha_t$ means the noise scheduling parameter at timestep t, while $\bar{\alpha}_t$ indicates the cumulative product of $\alpha$ from step 1 to t. Given $\tilde{z}_0$, we obtain highly correlated assets via the Decoder of VAE.

Naive noise interpolation based on a constant $\lambda$ and other diversity-inducing methods (e.g., entropy regularization, contrastive objective) are not robust to perturbation from $N_{Div}$. Our PROOF learns the adaptive interpolation weight based on the closed-form solution of OTIB. We define our noise finetuning as:

$$\theta^* = argmin_\theta \mathbb{E}_{N_{Orig}, N_{Div}}[\mathcal{L}_{noise}(PROOF_\theta(N_{Oirg}, N_{Div}), N_{Orig}) + \mathcal{L}_{info}(PROOF_\theta(N_{Orig}), \lambda)], \tag{3}$$

Figure 2: Method overview: as a plug-and-play content controller, PROOF can be employed for 2D/3D generation tasks, different architectures and model checkpoints. OTIB consists of a Sinkhorn Attention module and an information bottleneck module. We obtain $N_{PROOF}$ by information compression of $N_{Orig}$ and information modulation of $N_{Div}$. More details are introduced in Section 4.

where $PROOF_\theta$ is the generator of PROOF, $\mathcal{L}_{noise}$ aims to provide pixel-level regularization for structure and appearance alignment with $N_{Orig}$, and $\mathcal{L}_{info}$ explores controlling appropriate neural feature leakage with consideration of contextual preservation, which learns the minimal sufficient representation to avoid the diversity collapse.

## 3.2 INFORMATION BOTTLENECK REVISITING

Let's denote the original input data, the corresponding label, and compressed information by $X$, $Y$, and $Z$. The information compression principle Tishby & Zaslavsky (2015) is a trade-off between task-related information preservation and the minimal sufficient information compression, by maximizing the sharable information of $Z$ and $Y$ while minimizing that of $Z$ and $X$:

$$\max_Z \mathbb{I}(Y; Z) - \beta \mathbb{I}(X; Z), \tag{4}$$

where $\mathbb{I}$ means the mutual information and $\beta$ is a trade-off weight. Let $R$ denote the feature representations of $X$, and the information loss is formulated as:

$$\mathbb{I}(X; Z) \triangleq \mathbb{I}(R; Z) \triangleq \mathcal{D}_{KL}[p(Z|R) \| q(Z)], \tag{5}$$

where $q(Z)$ with Gaussian distribution is a variational approximation of $p(Z)$ Schulz et al. (2020). $\mathcal{D}_{KL}$ is the KL divergence Csiszár (1975) used to represent the distance between two distributions.

In our problem setting, $R$ is the noise latent $N_{Orig}$ and Z is the compressed latent $N_{Out}$.

## 3.3 OPTIMAL TRANSPORT REVISITING

We revisit the Optimal Transport that provides a mathematical framework for transporting probability distributions from the source to the target. Given discrete distributions as:

$$\mu = \sum_{i=1}^{M} \mu_i \delta_{x_i}, \quad \nu = \sum_{j=1}^{N} \nu_j \delta_{y_j} \tag{6}$$

where $\mu, \nu$ are discrete probability measures, $\mu_i \geq 0$, $\nu_j \geq 0$ are probability masses ($\sum_i \mu_i = \sum_j \nu_j = 1$), $\delta_x$ denotes the Dirac delta function centered at point $x$, $M$ and $N$ are the number of support points. The original OT problem finds a transport plan $\mathbf{T}^*$ that minimizes the total transportation cost, which is computationally intensive. The Sinkhorn algorithm Cuturi (2013); Kim et al. (2024) equips OT with an entropy regularization term:

$$\mathbf{T}^* = \arg \min_{\mathbf{T} \in \Pi(\mu, \nu)} \langle \mathbf{T}, \mathbf{C} \rangle_F - \epsilon H(\mathbf{T}), \tag{7}$$

where $\mathbf{T} \in \mathbb{R}^{M \times N}$ is the transport matrix with $\mathbf{T}_{ij}$ specifying how much mass moves from $x_i$ to $y_j$, $\mathbf{C} \in \mathbb{R}^{M \times N}$ is the cost matrix where $\mathbf{C}_{ij} = d(x_i, y_j)$, $\Pi(\mu, \nu) = \{\mathbf{T} \geq 0 \mid \mathbf{T1^N} =$

$\mu, \mathbf{T}^{\top}\mathbf{1^M} = \nu\}$ defines the set of admissible transport plans, $\langle \cdot, \cdot \rangle_F$ denotes the Frobenius inner product. Moreover, $\epsilon > 0$ is the regularization strength, $H(\mathbf{T}) = -\sum_{ij} \mathbf{T}_{ij} \log \mathbf{T}_{ij}$ is the entropy of the transport plan.

## 4 APPROACH

In this section, we provide a detailed introduction to our proposed PROOF, including the overall pipeline in Section 4.1, OTIB module architecture in Section 4.2, the closed-form theoretical solution in Section 4.3, along with the training loss in Section 4.4.

### 4.1 OVERALL PIPELINE

As shown in Fig. 2, PROOF can manipulate random noise with text or image conditions in 2D Rombach et al. (2022); Esser et al. (2024) or 3D data Xiang et al. (2025) distribution.

#### 4.1.1 PROOF_2D

As for none-referenced PROOF_2D, given a text prompt denoted by 'S', diverse images can be synthesized based on:

$$I_{PROOF} = G_\phi^{2D*}(PROOF_\theta^{2D}(N_{Oirg}, N_{Div}), \text{'S'}), \tag{8}$$

where $G_\phi^{2D*}$ is the frozen generator of diffusion model Rombach et al. (2022).

As for referenced PROOF_2D, given a reference image $I_{Ref}$, we extract the image prompt using IP-Adapter Ye et al. (2023) for consistent appearance transfer. Furthermore, we utilize the diffusion inversion method Mokady et al. (2023) to recover the corresponding contextual latent of $I_{Ref}$. $PROOF_\theta^{2D}$ perturbs the inversed noise to generate diverse images:

$$I_{PROOF} = G_\phi^{2D*}(PROOF_\theta^{2D}(Inv(I_{Ref}), N_{Div}), I_{Ref}) \tag{9}$$

#### 4.1.2 PROOF_3D

TRELLIS Xiang et al. (2025) compresses the 3D asset representation into a structured 3D latent similar to Latent Diffusion Rombach et al. (2022). It's possible for $PROOF_\theta^{3D}$ to implement the 3D tradeoff considering structural and textural preservation, along with the distribution diversity of 3D models and neural rendering Mildenhall et al. (2021); Kerbl et al. (2023); Lu et al. (2024):

$$M_{PROOF} = G_\phi^{3D*}(PROOF_\theta^{3D}(N_{Oirg}, N_{Div}), \text{'S'}), \tag{10}$$

where $G_\phi^{3D*}$ is the frozen generator of TRELLIS Xiang et al. (2025).

### 4.2 OTIB ARCHITECTURE

As mentioned in Section 3, implicit neural compression of information can be formulated as follows:

$$\min_Z \beta \mathbb{I}(N_{Orig}; Z), \tag{11}$$

where $\mathbb{I}$ denotes the mutual information function, $Z$ is the manipulated latent derived from $N_{Orig}$ via Equ. 1. To realize high-fidelity content preservation and generation diversity, we adaptively learn a neural information filter $\lambda$ of OTIB.

$$\lambda = Sigmoid(Conv(N_{Orig}) + \mathcal{F}_{SA}(N_{Orig})), \tag{12}$$

where $\mathcal{F}_{SA}$ is a Sinkhorn Attention module, as shown in Figure 2. The intent of PROOF is to improve representation diversity while implicitly adhering to the global content attributes of a certain scenario. If $\lambda$ is 0, the whole manifold will be replaced by $N_{Div}$, which results in entire structure and appearance leakages. If $\lambda$ is 1, $Z$ excludes any form of diversity-inducing perturbations, which results in mode collapse. Qualitative analyses are illustrated in Sec. 5.

### 4.3 CLOSED-FORM SINKHORN-IB SOLUTION

We impose a Sinkhorn Attention module $\mathcal{F}_{SA}$ in a spatial-OT view to improve contextual preservation of PROOF. The Sinkhorn Attention algorithm is as follows:

---

**Algorithm 1** Sinkhorn-Attention Forward Pass

---
1: **Input:** Feature map $X \in \mathbb{R}^{B \times C \times H \times W}$
2: $Q = \text{Conv\_Nd}(X)$, $K = \text{Conv\_Nd}(X)$, $V = \text{Conv\_Nd}(X)$        ▷ Learnable projections
3: $A = QK^{\top}/\sqrt{C}$        ▷ Attention logits
4: **for** $k = 1$ to $n_{iters}$ **do**
5:     $A = A - \text{LogSumExp}(A, \dim = 2)$        ▷ Row normalization
6:     $A = A - \text{LogSumExp}(A, \dim = 1)$        ▷ Column normalization
7: **end for**
8: $\mathbf{T} = \exp(A)$        ▷ Optimal attention weights
9: **return** $\mathbf{T}V$        ▷ Transport applied to values

---

where $Q, K, V \in \mathbb{R}^{B \times (HW) \times C}$ are Query, Key, Value tensors, respectively. $A \in \mathbb{R}^{B \times (HW) \times (HW)}$ is Attention logits matrix, $\text{LogSumExp}(A)_i = \log \sum_j \exp(A_{ij})$, and $\mathbf{T}$ is Doubly-stochastic attention matrix. Our transport solution is established through:

$$\mathbf{T}_{ij} = \exp(\underbrace{\frac{q_i^{\top} k_j}{\sqrt{C}}}_{\text{Transport cost}} - \underbrace{\alpha_{OT}^i - \beta_{OT}^j}_{\text{Sinkhorn scalars}}) \tag{13}$$

where $\alpha_{OT}$ and $\beta_{OT}$ are row and column normalization factors, respectively. The division by $\sqrt{C}$ stabilizes gradient flow. We consider the joint optimization objective of OTIB:

$$\min_{\lambda} \underbrace{I(R; Z)}_{\text{IB term}} + \gamma \underbrace{< A^*, \mathbf{C} >}_{\text{Sinkhorn OT term}} + \epsilon H(A^*), \tag{14}$$

where $Z = \lambda \odot N_{Orig} + (1 - \lambda) \odot N_{Div}$, $A^* = \text{Sinkhorn}(\mathbf{C})$, where $\mathbf{C}_{ij} = \frac{\langle q_i, k_j \rangle}{\sqrt{d}}$, $d = C$.

We assume that: $N_{Orig} \sim \mathcal{N}(0, \sigma_R^2 I)$, $N_{Div} \sim \mathcal{N}(0, \sigma_{N_{Div}}^2 I)$. $N_{Orig}$ and $N_{Div}$ are independent.

**Step 1**: Information Bottleneck Term Simplification. Under Gaussian assumptions, the mutual information and the gradient calculation are formulated as:

$$I(R; Z) = \frac{1}{2} \log \left( 1 + \frac{\lambda^2 \sigma_R^2}{(1 - \lambda)^2 \sigma_{N_{Div}}^2} \right), \frac{\partial I}{\partial \lambda} = \frac{\lambda \sigma_R^2 - (1 - \lambda) \sigma_{N_{Div}}^2}{\lambda^2 \sigma_R^2 + (1 - \lambda)^2 \sigma_{N_{Div}}^2} \tag{15}$$

**Step 2**: Sinkhorn Term Gradient. Using the Envelope Theorem and chain rule:

$$\frac{\partial \mathcal{L}_{OT}}{\partial \lambda} = \left\langle \frac{\partial A^*}{\partial \lambda}, \mathbf{C} \right\rangle + \left\langle A^*, \frac{\partial \mathbf{C}}{\partial \lambda} \right\rangle \approx \left\langle A^*, \frac{\partial \mathbf{C}}{\partial \lambda} \right\rangle, \tag{16}$$

where $A^* = \text{diag}(u) K \text{diag}(v)$ with $K = e^{-\mathbf{C}/\epsilon}$. $\frac{\partial \mathbf{C}_{ij}}{\partial \lambda} = \frac{\partial}{\partial \lambda}(\frac{<q_i, k_j>}{\sqrt{d}}) = \frac{1}{\sqrt{d}} \langle q_i, \frac{\partial k_j}{\partial Z_j} \cdot \frac{\partial Z_j}{\partial \lambda} \rangle \approx \frac{1}{\sqrt{d}} \langle q_i, \frac{\partial k_j}{\partial N_{Orig}^j} \cdot \frac{\partial Z_j}{\partial \lambda} \rangle = \frac{1}{\sqrt{d}} \langle q_i, W_K (N_{Orig}^j - N_{Div}^j) \rangle$.

**Step 3**: First-Order Optimality Condition Setting. The total gradient to zero:

$$\frac{\lambda \sigma_R^2 - (1 - \lambda) \sigma_{N_{Div}}^2}{\lambda^2 \sigma_R^2 + (1 - \lambda)^2 \sigma_{N_{Div}}^2} + \gamma \mathbb{E}_{A^*} \left[ \frac{\partial \mathbf{C}_{ij}}{\partial \lambda} \right] = 0 \tag{17}$$

**Step 4**: Closed-Form OTIB Solution. The optimal weights take the form (More details are in Appendix A):

$$\lambda^* = \sigma \left( \frac{1}{\eta} \left( \gamma \cdot \text{Align}(N_{Orig}, N_{Div}) - \frac{\sigma_{N_{Div}}^2}{\sigma_R^2} \right) \right), \tag{18}$$

where $\text{Align}(\cdot) = \mathbb{E}_{A^*} \left[ \frac{\partial \mathbf{C}_{ij}}{\partial \lambda} \right]$, $\sigma(\cdot)$ is the sigmoid function, and $\eta$ is a hyperparameter. The closed-form solution is aligned with Equ. 12, where $Conv$ approximates $\sigma^2$ ratio, and $\mathcal{F}_{SA}$ approximates Align term.

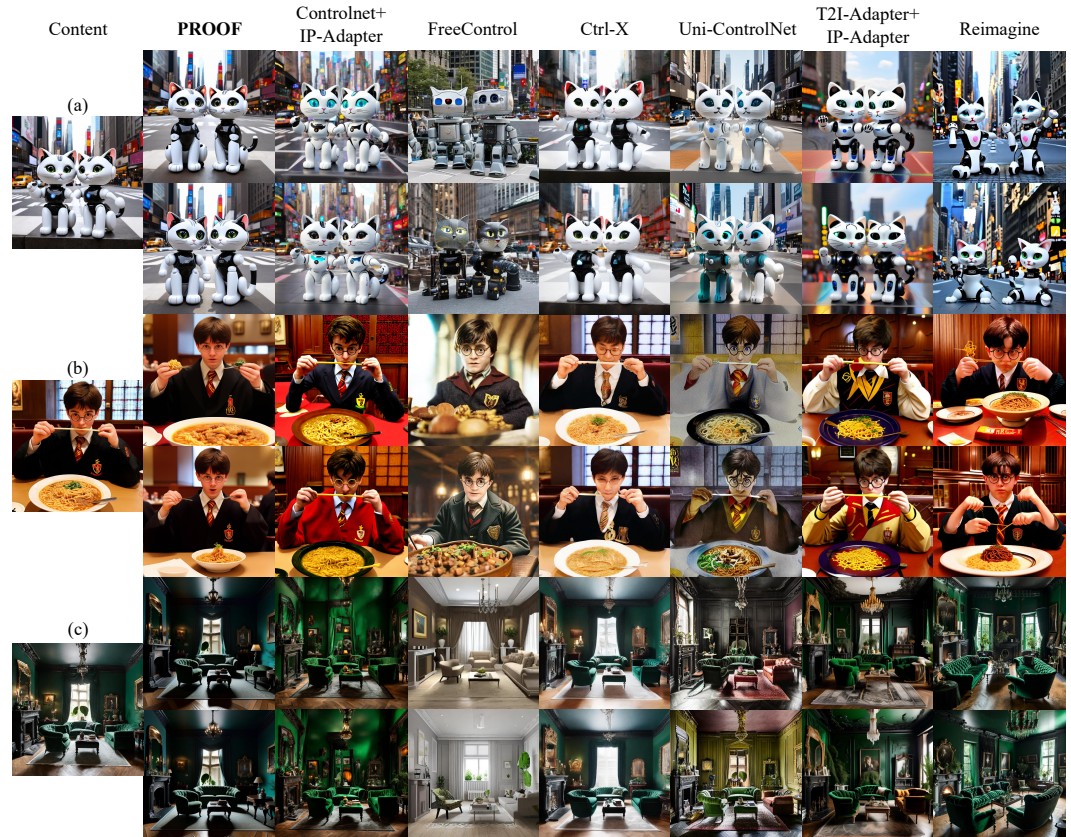

| Content | **PROOF** | Controlnet+ IP-Adapter | FreeControl | Ctrl-X | Uni-ControlNet | T2I-Adapter+ IP-Adapter | Reimagine |

Figure 3: Qualitative results of PROOF_2D, ControlNet + IP Adapter Zhang et al. (2023a); Ye et al. (2023), FreeControl Mo et al. (2024), Ctrl-X Lin et al. (2024), Uni-ControlNet Zhao et al. (2023), T2I-Adapter + IP Adapter Mou et al. (2024); Ye et al. (2023), and Reimagine AI (2023). Zoom in for better observation. PROOF realizes more controllable image variations with high-fidelity content.

## 4.4 TRAINING LOSS

Training losses contain pixel-level reconstruction loss and manifold-level information compression loss. As for noise consistency loss, the pixel-level supervision for $N_{PROOF}$ is MSE loss that demonstrates a powerful content preservation function Rombach et al. (2022); Ruiz et al. (2023):

$$\mathcal{L}_{noise} = ||N_{PROOF} - N_{Orig}||_2^2. \tag{19}$$

For Gaussian distribution $\mathcal{N}(\mu, \sigma^2)$ and $\mathcal{N}(0, 1)$, KL divergence is formulated as:

$$\mathcal{D}_{KL}[N(\mu, \sigma^2)||N(0, 1)] = -\frac{1}{2}[log(\sigma)^2 - (\sigma)^2 - (\mu)^2 + 1]. \tag{20}$$

Our framework eliminates the need for feature mean/variance pre-calculation by leveraging the predefined properties of Gaussian noise ($\mu_G=0$, $\sigma_G=1$). As for our case mentioned in Equ. 5, the distribution of $p(Z|R)$ is accessed as $\mathcal{N}[\lambda R, (1 - \lambda)^2]$ according to Equ. 1. We normalize $p(Z|R)$ along with $q(Z)$ using $\mu_G$ and $\sigma_G$, then the information compression metric of PROOF is:

$$\mathcal{L}_{info} = \mathbb{I}(Z; R) = KL[p(Z|R)||q(Z)] = -\frac{1}{2}[log(1 - \lambda)^2 - (1 - \lambda)^2 - (\lambda R)^2 + 1], \tag{21}$$

Finally, the total loss of PROOF is formulated as:

$$\mathcal{L}_{PROOF} = \beta \mathcal{L}_{info} + \mathcal{L}_{noise}, \tag{22}$$

where $\beta$ is the content-diversity tradeoff weight (Fig. 10a). Higher $\beta$ usually intentionally relaxes contextual constraints but boosts the diversity (Fig. 13, Fig. 19).

Table 1: PROOF outperforms other SOTA methods in structure and appearance alignments and robustness, measured by DINO ViT self-similarity and DINO-I. We report the inference time of PROOF_2D_Ref, where diffusion inversion Mokady et al. (2023) is time-consuming. We assess image quality (PickScore, HPSv2, AES) and diversity (LPIPS, L1).

| Methods | Training | Inference time (s) | self-sim ↓ | DINO-I ↑ | PickScore↑ | HPSv2↑ | AES↑ | L1 | LPIPS |
|---|---|---|---|---|---|---|---|---|---|
| Uni-ControlNet Zhao et al. (2023) | ✔ | 10.6 | 0.045 | 0.555 | 6.49 | 25.33 | 6.26 | 56.41 | 0.5500 |
| ControlNet + IP Adapter Zhang et al. (2023a) | ✔ | 8.1 | 0.068 | 0.656 | 15.08 | 25.02 | 6.29 | 46.06 | 0.4334 |
| T2I-Adapter + IP Adapter Mou et al. (2024) | ✔ | 4.2 | 0.055 | 0.603 | 12.39 | 25.45 | 6.28 | 50.45 | 0.4436 |
| Ctrl-X Lin et al. (2024) | ✘ | 14.9 | 0.057 | 0.686 | 11.65 | 24.63 | 6.27 | 37.07 | 0.4812 |
| FreeControl Mo et al. (2024) | ✘ | 21.5 | 0.058 | 0.572 | 18.13 | 26.13 | 6.19 | **85.45** | **0.636** |
| Reimagine AI (2023) | ✔ | 10.1 | 0.073 | 0.753 | 15.14 | 25.27 | 6.34 | 64.12 | 0.6192 |
| RIVAL Zhang et al. (2023b) | ✘ | 13.91 | 0.035 | 0.826 | 56.64 | 21.12 | 6.22 | 47.50 | 0.5431 |
| Prompt-Free Diffusion Xu et al. (2024) | ✔ | 10.91 | 0.025 | 0.824 | 22.35 | 19.92 | 6.21 | 40.36 | 0.4671 |
| **PROOF (ours)** | ✔ | 27.2 | 0.038 | 0.841 | 16.61 | 25.67 | 6.29 | 41.58 | 0.4342 |

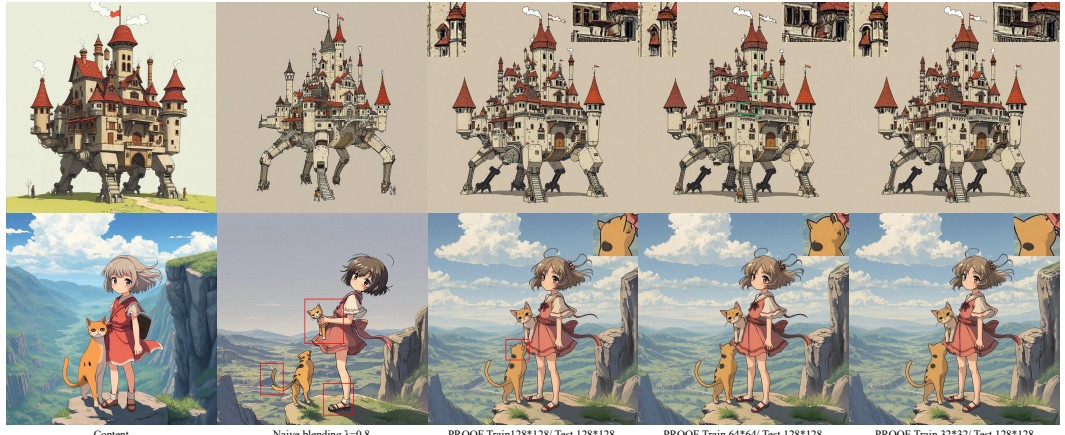

| Content | Naive blending λ=0.8 | PROOF Train128*128/ Test 128*128 | PROOF Train 64*64/ Test 128*128 | PROOF Train 32*32/ Test 128*128 |

Figure 4: Robust inference performance of PROOF across distinct latent resolutions. We set $\beta$ of PROOF as 0.2, which is aligned with $\lambda = 0.8$. It's efficient for Sinkhorn attention and information bottleneck to finetune on low-resolution noise space while inferring on high-resolution latent.

## 5 EXPERIMENTS

Comprehensive qualitative and quantitative evaluations validate PROOF's dual capability in maintaining content fidelity while enhancing generation diversity for digital asset creation. Training protocol and baselines are presented in Appendix B and C. Additional results, e.g., golden noise Zhou et al. (2025) finetune (Fig. 18), are shown in Appendix F.

### 5.1 QUANTITATIVE EVALUATION

Tab. 1 shows a quantitative comparison of natural images of datasets Lin et al. (2024). The content alignment metrics include DINO ViT self-similarity Tumanyan et al. (2022), DINO-I Ruiz et al. (2023) (details are explained in Appendix D). Note that PROOF shows consistent superiority on self-sim and DINO-I scores. As for image quality, we utilize PickScore Kirstain et al. (2023), HPSv2 Wu et al. (2023), and Aesthetic Score (AES) Schuhmann (2023). We assess the diversity via LPIPS Zhang et al. (2018) between the source image and the generated image. Note that PROOF_2D using SD-3 takes around 7s, which is more efficient for variant generation. Meanwhile, the subjective metrics consist of quality, fidelity, and diversity without compromising fidelity. PROOF achieves comparable user preference (Tab. 4).

### 5.2 QUALITATIVE RESULT

PROOF only learn noise representation supervised by itself based on OTIB. Visually comparable results demonstrate that our implicit PROOF is a better workbench for highly correlated asset editing (Fig. 3, Fig. 18, more examples in Appendix F), based on robust noise representation learning.

**Content transformation** Although PROOF applies intrinsic interpolation to manipulate noise, the latent space compressed by VAE is already a high-dimensional manifold where nonlinear content transformations are represented to some extent. That means the change of a specific noise point with a certain channel and position is capable of imposing contextual transformation on several image-level areas, therefore leading to visually object deformation or novel-view rendering (Fig. 14).

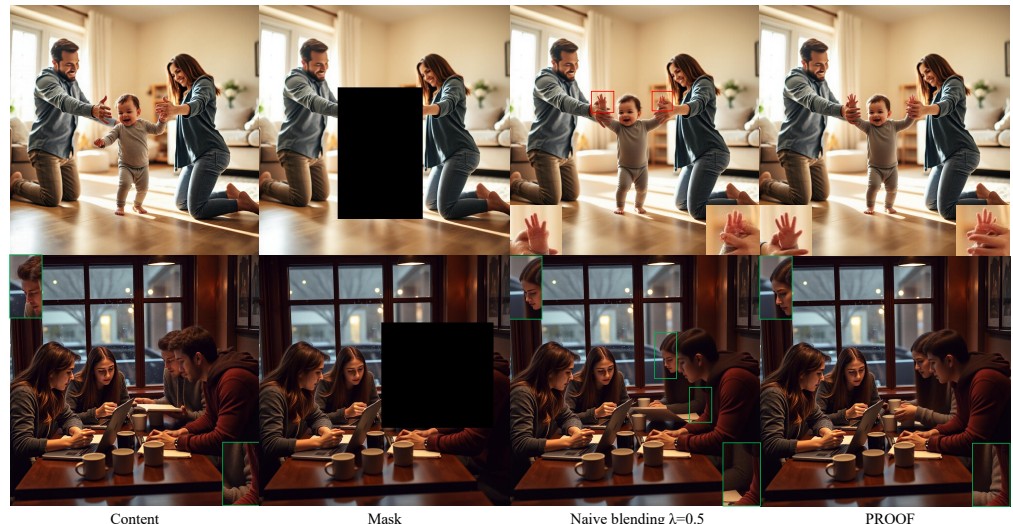

Figure 5: Robust local editing visualization. PROOF preserves local content layout and synthesizes controllable and diverse inpainting results with highly faithful details.

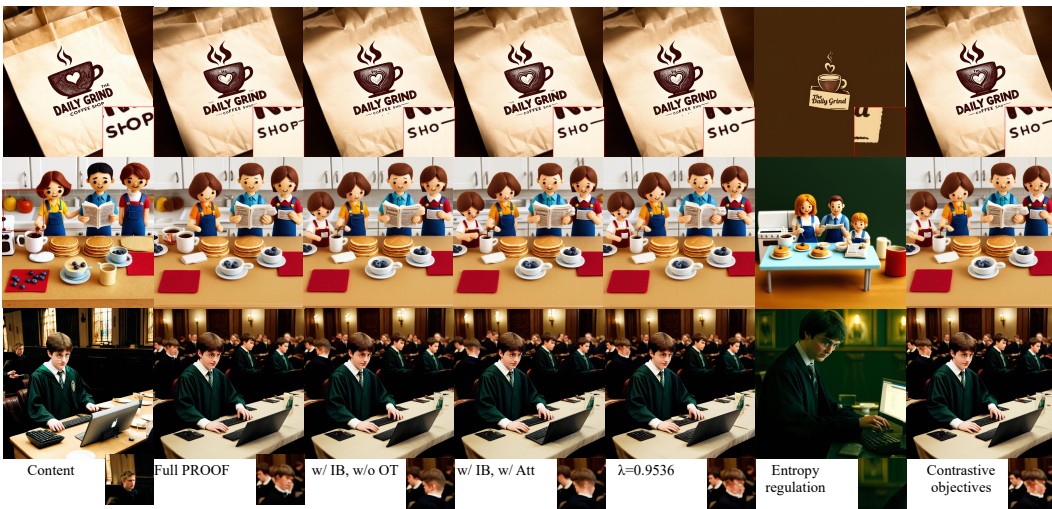

Figure 6: PROOF sufficiently preserves the global structure and appearance based on OTIB, e.g., the word 'SHOP', no-man's land on the left of Row 2, and the far-distance face of Row 3, while other variants show lower content fidelity. More results are illustrated in Fig. 8.

**Train-Test resolution discrepancy**  We conduct experiments concerning the latent resolution discrepancy between the fine-tuning and inference phases (Fig. 4). The overall contents of different finetuning models are consistent. However, the finetuning model employing 32-resolution data (Col 5) hardly captures local topological and textural details when dealing with 128-resolution inference.

**Local variation**  PROOF can be employed by generation models equipped with the inpainting function to implement local content variation. As shown in Fig. 5, it's also important to provide uniform attention distribution based on optimal transport in the local editing scenario. PROOF synthesizes higher-fidelity and higher-quality human components. Furthermore, we evaluate PROOF on the edge controller (Fig. 11) and semantic editing (Fig. 12), which significantly strengthens PROOF's generalizability to broader applications.

**DiT-based model generalization**  Whether PROOF can be applied to more advanced diffusion models featuring distinct architectural frameworks, e.g., Flux or SD3.5 based on Diffusion Transformer, has been further investigated. Conducting empirical validations on such state-of-the-art models substantially reinforces PROOF's ability to generalize and amplify its broader applicability across scenarios. Note that Figures 4, 5, 7 are all based on Flux Labs et al. (2025). Additionally, Fig. 17 shows robust variant results using SD-3.5 Esser et al. (2024). We report the computational complexity comparison of OTIB for different models in Tab. 3.

Table 2: Quantitative validation for PROOF_2D generation with random noise initialization on the dataset Lin et al. (2024). PROOF outperforms other ablation configurations and diversity-inducing methods in structure and appearance alignments. $w$ is the loss weight aligned with $\lambda$.

| Configuration | self-sim ↓ | DINO-I ↑ | PickScore↑ | HPSv2↑ | AES↑ | LPIPS |
|---|---|---|---|---|---|---|
| w/o IB ≜ Content | 0 | 0.9999 | 20.95 | 34.64 | 5.50 | 0 |
| Full PROOF $\beta$=0.05 | **0.0314** | **0.9026** | 18.43 | 33.38 | 5.43 | 0.4551 |
| w/ IB, w/o OT | 0.0333 | 0.8974 | 18.20 | 33.35 | 5.36 | 0.4562 |
| w/ IB, w/ AttentionBlock | 0.0331 | 0.8968 | **18.81** | **33.80** | 5.37 | 0.4590 |
| Naïve interpolation $\lambda$=0.9536 | 0.0423 | 0.8650 | 14.85 | 33.15 | 5.39 | 0.4549 |
| Entropy regularization $w$=0.45 | 0.0947 | 0.6299 | 12.30 | 31.20 | **5.76** | **0.6790** |
| Contrast objective $w$=0.085 | 0.0320 | 0.9012 | 17.38 | 33.45 | 5.41 | 0.4565 |

Table 3: Computational complexity comparison of OTIB for different models.

| Models | Spatial latent | MACs | Params | Inference time |
|---|---|---|---|---|
| SD-1.4, SD-1.5, SD-2 | (B, 4, 64, 64) | 134.64 MMac | 100 | 0.1579s |
| SD-3, SD-3.5, Flux | (B, 16, 128, 128) | 8.61 GMac | 1.36 k | 0.2185s |

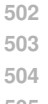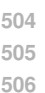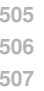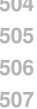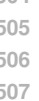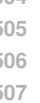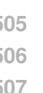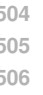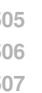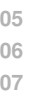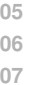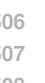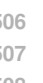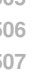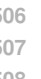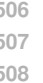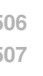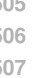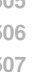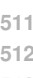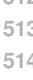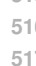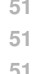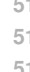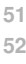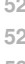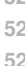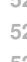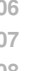

Content | $\lambda$=0.99 | $\lambda$=0.95 | $\lambda$=0.9 | $\lambda$=0.8 | $\lambda$=0.7 | $\lambda$=0.6 | $\lambda$=0.5 (Naïve blending)

$\beta$=0.01 | $\beta$=0.053 | $\beta$=0.105 | $\beta$=0.2 | $\beta$=0.3 | $\beta$=0.414 | $\beta$=0.56 (PROOF)

Figure 7: Comparison of naive blending and PROOF over a wide parameter range. Naïve blending method leads variants with lower $\lambda$ to suffer from significant structure and appearance distortions. Whereas PROOF preserves content more robustly under strong perturbation injection.

## 5.3 ABLATION STUDY

Fig. 6 demonstrates substantial benefits of PROOF over other alternatives, e.g., naive interpolation, entropy regularization, contrastive objective (loss details in Appendix E). PROOF exhibits best structure and appearance fidelity, and comparative perceptual quality in Tab. 2. Without the Information Bottleneck, the model will suffer from mode collapse due to a lack of mode diversity. Moreover, as shown in Fig. 10 (a), the PROOF variants without Sinkhorn Attention fail to capture local structure and appearance patterns (red boxes in col 3&4).

**Tradeoff weight** $\beta$    The context-diversity tradeoff weight $\beta$ controls the structure and appearance leakage in an adaptive way (Fig. 19). As illustrated in Fig. 7, lower $\lambda$ brings up more changes for variants. Nevertheless, PROOF exhibits better perturbation robustness compared with naive blending, which demonstrates the effectiveness of the closed-form solution and architecture of OTIB.

## 5.4 LIMITATIONS

Large-scale compression with small weight $\lambda$ may result in background leakage to a certain extent, as shown in Figure 10 (b). Nevertheless, the pose and identity of the original content are preserved as much as possible.

## 6 CONCLUSION

Our proposed PROOF conducts perturbation-robust asset creation with a trade-off of fidelity and diversity. We derive the closed-form solution of the optimal transported information bottleneck and design an efficient and effective OTIB module. Compared with explicit content alignment methods, along with other diversity-inducing alternatives, PROOF preserves topology and texture better. Comprehensive experimental analyses demonstrate that PROOF is promising to be the first plug-and-play implicit controller for pre-trained conditional 2D/3D generation models with remarkable context consistency and controllable diversity.

**Broader impacts.**    Our method provides a robust editor for both images and 3D models. While its primary advantage lies in assisting designers, animators, and 3D modelers in asset creation, the potential for malicious manipulation of visual assets necessitates mandatory watermarking in practical applications.

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

## A   APPENDIX A: DETAILED DERIVATION OF CLOSED-FORM SOLUTION

1. **Initial optimality condition:** Based on step 3 of Section 4.3, the optimization problem of OTIB gives us:

$$\frac{\lambda\sigma_R^2 - (1-\lambda)\sigma_{N_{Div}}^2}{\lambda^2\sigma_R^2 + (1-\lambda)^2\sigma_{N_{Div}}^2} + \gamma\text{Align} = 0 \tag{23}$$

This equation balances the information bottleneck term with the optimal transport term.

2. **Rearrange optimality condition:** We multiply both sides by the denominator to eliminate the fraction:

$$\lambda\sigma_R^2 - (1-\lambda)\sigma_{N_{Div}}^2 = -\gamma\text{Align}(\lambda^2\sigma_R^2 + (1-\lambda)^2\sigma_{N_{Div}}^2) \tag{24}$$

This form removes the denominator but introduces quadratic terms in $\lambda$.

3. **Auxiliary function definition:** To analyze this equation, we define:

$$f(\lambda) = \lambda\sigma_R^2 - (1-\lambda)\sigma_{N_{Div}}^2 + \gamma\text{Align}\left[\lambda^2\sigma_R^2 + (1-\lambda)^2\sigma_{N_{Div}}^2\right] \tag{25}$$

The optimal solution occurs when $f(\lambda) = 0$.

4. **Taylor expansion at $\lambda = 0.5$:** We linearize around $\lambda = 0.5$ because:

- It's the midpoint of possible $\lambda$ values
- The function is most linear in this region
- Higher-order terms are minimized here

**4.1. Function value at $\lambda = 0.5$:**

$$f(0.5) = 0.5(\sigma_R^2 - \sigma_{N_{Div}}^2) + 0.25\gamma\text{Align}(\sigma_R^2 + \sigma_{N_{Div}}^2) \tag{26}$$

This combines the linear difference and quadratic alignment terms.

**4.2. First derivative:**

$$f'(\lambda) = \sigma_R^2 + \sigma_{N_{Div}}^2 + \gamma\text{Align}\left[2\lambda\sigma_R^2 - 2(1-\lambda)\sigma_{N_{Div}}^2\right] \tag{27}$$

$$f'(0.5) = \sigma_R^2 + \sigma_{N_{Div}}^2 + \gamma\text{Align}(\sigma_R^2 - \sigma_{N_{Div}}^2) \tag{28}$$

The derivative shows how sensitive the function is to $\lambda$ changes.

**4.3. Linear approximation solution:** Using Taylor expansion:

$$\lambda \approx 0.5 - \frac{f(0.5)}{f'(0.5)} = 0.5 - \frac{0.5(\sigma_R^2 - \sigma_{N_{Div}}^2) + 0.25\gamma\text{Align}(\sigma_R^2 + \sigma_{N_{Div}}^2)}{\sigma_R^2 + \sigma_{N_{Div}}^2 + \gamma\text{Align}(\sigma_R^2 - \sigma_{N_{Div}}^2)} \tag{29}$$

This gives us a first-order approximation of the optimal $\lambda$.

**4.4. Simplified linear expression:** When $\gamma\text{Align}$ is relatively small compared to the variance terms:

$$\lambda \approx \underbrace{\frac{\sigma_{N_{Div}}^2}{\sigma_R^2 + \sigma_{N_{Div}}^2}}_{C} + \underbrace{0.25\gamma}_{K}\cdot\text{Align}, \tag{30}$$

where $C$ represents the baseline compression ratio, and $K$ determines how strongly alignment affects the result.

5. **Identify limitations of the linear form:** The linear expression has two critical flaws:

- When Align is too large, $\lambda$ may exceed 1
- When Align is too small, $\lambda$ may be less than 0

However, $\lambda$ must be a weight coefficient strictly between 0 and 1. Therefore, we need a function that constrains the output to (0,1) while preserving the positive correlation between $\lambda$ and Align.

6. **Choose sigmoid function for constraint:** The sigmoid function $\sigma(x) = \frac{1}{1+e^{-x}}$ is ideal because:

- Its output is strictly bounded between (0,1)
- It's monotonically increasing, preserving the positive correlation
- It provides smooth, differentiable transitions

7. **Match the baseline value at Align $= 0$:** When there's no alignment (Align $= 0$), the linear expression gives $\lambda \approx C$. To maintain consistency:

$$\sigma(x_0) = C \quad \text{where } x_0 = \sigma^{-1}(C) \tag{31}$$

Using the inverse of the sigmoid function (logit function) $\sigma^{-1}(y) = \ln\left(\frac{y}{1-y}\right)$, we get:

$$\sigma^{-1}(C) = \ln\left(\frac{\sigma_{N_{Div}}^2}{\sigma_R^2}\right) \tag{32}$$

This ensures the sigmoid preserves the baseline behavior when Align $= 0$.

8. **Final sigmoid parameterization:** To maintain the positive correlation while adding flexibility, we introduce:

$$x = \frac{1}{\eta}\left(\gamma \cdot \text{Align} - \frac{\sigma_{N_{Div}}^2}{\sigma_R^2}\right), \tag{33}$$

where $\eta > 0$ controls the steepness of the transition. The final solution becomes:

$$\lambda^* = \sigma\left(\frac{1}{\eta}\left(\gamma \cdot \text{Align} - \frac{\sigma_{N_{Div}}^2}{\sigma_R^2}\right)\right) \tag{34}$$

This closed-form solution is presented in step 4 of Section 4.3, and satisfies all our requirements:

- Strictly bounded output (0,1)
- Preserves positive correlation
- Matches baseline when Align $= 0$
- Allows tuning via $\eta$ and $\gamma$

## B  TRAINING PROTOCOL

We train our PROOF on Gaussian noise tensors with corresponding dimension shape of different architectures, e.g., $4*64*64$ Rombach et al. (2022), $16*128*128$ Esser et al. (2024), $8*16*16*16$ Xiang et al. (2025). $N_{Orig}$ and $N_{Div}$ are random noises in each training step. As for PROOF_3D, we utilize 3D convolutions for SA and IB modules. We train PROOF for 20k iterations with one NVIDIA RTX 4090 GPU. The training batch size is set to 1. During training, we employ Adam Kingma & Ba (2014) with $2*10^{-3}$ learning rate. We set $\beta = 0.01$ for mild diversity (Figure 3a), $\beta = 0.1$ for substantial diversity (Fig. 3b, Fig. 13), and $\beta = 1$ for diversity with reference constraints (Fig. 3c).

## C  BASELINES

There are several state-of-the-art controllable synthesis methods based on diffusion models. ControlNet Zhang et al. (2023a) and T2I-Adapter Mou et al. (2024) align diffusion priors to the external control structures. We further apply IP-Adapter Ye et al. (2023) to them for better textural transfer. These methods present low topological flexibility with restriction by the explicit structure alignment, and limited textural fidelity with global appearance control. FreeControl Mo et al. (2024) has large-scale content variance due to imprecise structure and appearance representations (col 4 in Fig. 3). Ctrl-X Lin et al. (2024) provides too-strict structure and appearance alignments, and there are texture distortions. Uni-ControlNet Zhao et al. (2023) also suffers from the global appearance representation (col 6 in Fig. 3). Reimagine AI (2023) produces uncontrollable content layout, despite high image quality and diversity (col 8 in Fig. 3). RIVAL Zhang et al. (2023b) conducts distribution alignment between the generative and inverse paths to realize semantic and structural fidelity. Prompt-free Diffusion Xu et al. (2024) discards the text encoder and text prompts, which may bring about semantic degradation. We evaluate all methods on SDXL v1.0 Podell et al. (2024) when workable and on their pre-configured base models otherwise.

810

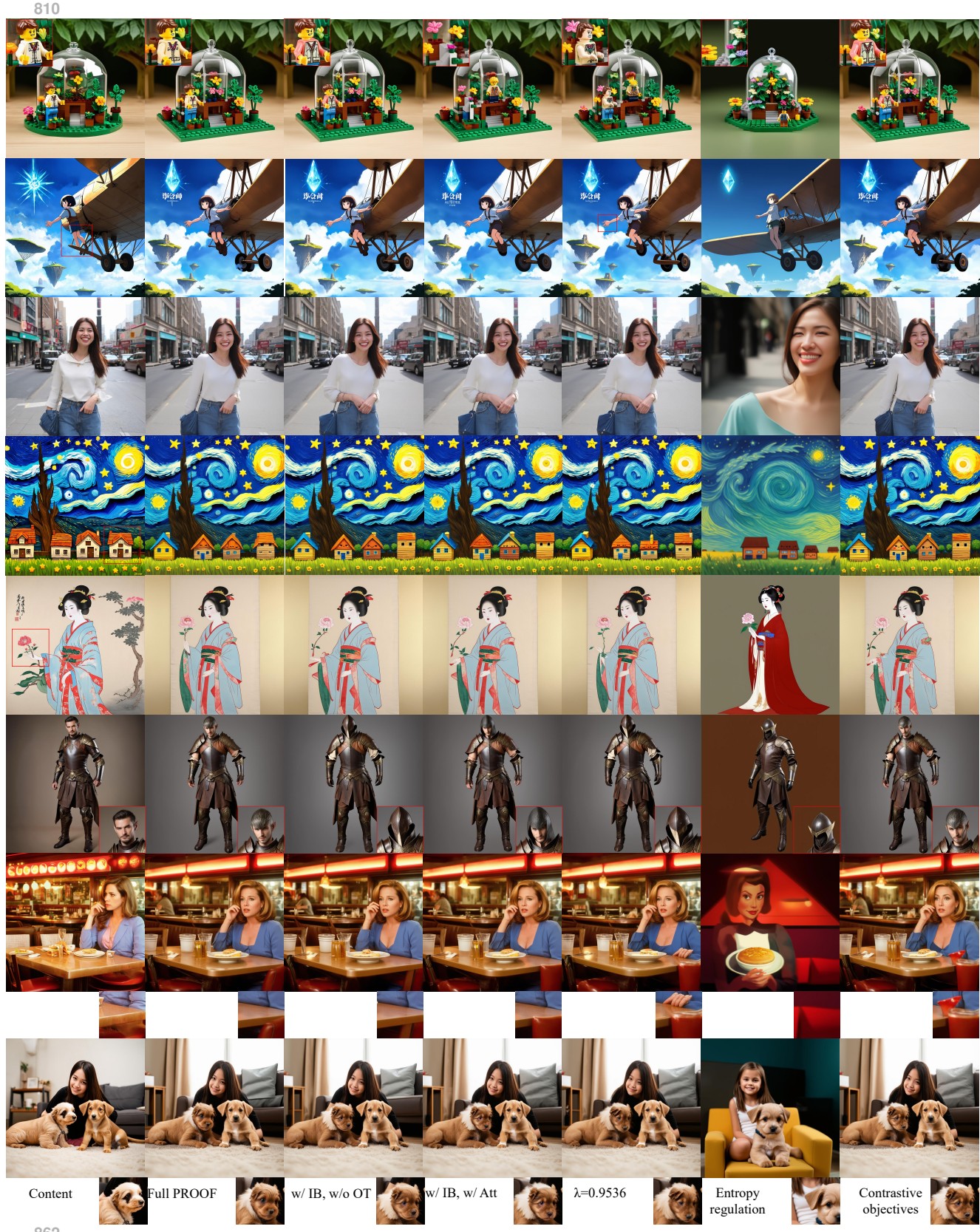

Figure 8: PROOF sufficiently preserves the global structure and appearance based on OTIB, while other variants show lower content fidelity. Zoom in for better observation.

862

863

## D  EVALUATION METRIC

Below is the explicit explanation of how DINO ViT self-similarity and DINO-I are calculated:

1. The structural consistency is quantified as:

$$\text{Self-sim} = \frac{1}{N} \sum_{i=1}^{N} \|\phi_{\text{DINO}}(I_{\text{Ref}})_i - \phi_{\text{DINO}}(I_{\text{Out}})_i\|_2^2, \tag{35}$$

where $\phi_{\text{DINO}}$: DINO-ViT base model (patch size=8) feature extractor, $I_{\text{Ref}}$: Reference input image, $I_{\text{Out}}$: Generated output image, $N$: Number of feature vectors (layer_num=11).

2. The appearance similarity is computed as:

$$\text{DINO-I} = \frac{\mathbf{v}_{\text{ref}} \cdot \mathbf{v}_{\text{out}}}{\|\mathbf{v}_{\text{ref}}\|_2 \|\mathbf{v}_{\text{out}}\|_2}, \tag{36}$$

where $\mathbf{v}_{\text{ref}} = \phi_{\text{DINO}}^{[\text{CLS}]}(I_{\text{ref}})$: DINO-ViT [CLS] token embedding of reference image, $\mathbf{v}_{\text{out}} = \phi_{\text{DINO}}^{[\text{CLS}]}(I_{\text{out}})$: DINO-ViT [CLS] token embedding of output image, $\phi_{\text{DINO}}$: DINO-ViT small model (patch size=16) feature extractor, '$\cdot$' denotes dot product.

## E  DIVERSITY-BOOSTING METHODS

In these diversity-inducing settings, we maintain the $\mathcal{L}_{\text{noise}}$ of Equ. 22 to conduct global content preservation.

### E.1  CONTRASTIVE OBJECTIVE

Given flat_Z = flatten$(Z) \in \mathbb{R}^{N \times d}$, flat_h = flatten$(R) \in \mathbb{R}^{N \times d}$, flat_l = flatten$(N_{Div}) \in \mathbb{R}^{N \times d}$, we calculate the cross-modal cosine similarity explicitly as sim_zh = cos(flat_Z, flat_h), sim_zl = cos(flat_Z, flat_l), sim_hl = cos(flat_h, flat_l). Then the contrastive objective loss is indicated as:

$$\mathcal{L}_{\text{contrast}} = w * (\text{MSE}(\text{sim\_zh}, \text{sim\_hl}) + \text{MSE}(\text{sim\_zl}, 1 - \text{sim\_hl})), \tag{37}$$

where $w$ is the loss weight.

Note that the contrastive objective has some limitations as follows:

1. Exhibits significantly weaker robustness compared to PROOF under strong perturbations.

2. Fails to perform effective representation learning at the manifold distribution level.

3. Demonstrates notable scalability constraints in real-world applications.

4. Generates structural and appearance artifacts (Fig. 6, Fig. 8).

### E.2  ENTROPY REGULARIZATION

1. Input tensor flattening (flatten the i-th sample of Z from multi-dimensional to a vector)

$$Z_i^{\text{flat}} = \text{view}(Z_i, -1) \tag{38}$$

(i.e., flattened into a $1 \times D$ vector, where $D$ is the flattened dimension)

2. Softmax probability calculation of the j-th class for the i-th sample (compute class probabilities for each flattened sample)

$$p_{i,j} = \text{Softmax}(Z_i^{\text{flat}})_j = \frac{\exp((Z_i^{\text{flat}})_j)}{\sum_{k=1}^{D} \exp((Z_i^{\text{flat}})_k)} \tag{39}$$

3. Entropy calculation for a single sample ($\epsilon$ is added to avoid meaningless logarithm)

$$H(Z_i) = -\sum_{j=1}^{D} p_{i,j} \cdot \log(p_{i,j} + \epsilon) \tag{40}$$

Table 4: PROOF exhibits competitive human preference percentages. Preference consistency is 87%, std. deviation is ±3.0%, and the p-value of Wilcoxon is 0.016, which demonstrates the results are statistically significant.

| Methods | Quality ↑ | Fidelity ↑ | Diversity (subject to Fidelity)↑ |
|---|---|---|---|
| Uni-ControlNet Zhao et al. (2023) | 78% | 71% | 75% |
| ControlNet + IP Adapter Zhang et al. (2023a); Ye et al. (2023) | 57% | 64% | 75% |
| T2I-Adapter + IP Adapter Mou et al. (2024); Ye et al. (2023) | 67% | 65% | 78% |
| Ctrl-X Lin et al. (2024) | 81% | **90%** | 74% |
| FreeControl Mo et al. (2024) | 76% | 51% | 66% |
| Reimagine AI (2023) | **91%** | 37% | 51% |
| **PROOF (ours)** | 89% | 89% | **90%** |

Therefore, the function definition of Entropy regularization is:

$$\text{Entropy}(Z, w, \epsilon = 10^{-8}) = -w \cdot \frac{1}{N} \sum_{i=1}^{N} H(Z_i) \tag{41}$$

Note that entropy regularization has some limitations as follows (Fig. 6, Fig. 8):

1. Complete loss of background information.

2. Fails to ensure a minimal sufficient representation learning.

3. Poor robustness in structure and appearance preservation.

# F   ADDITIONAL RESULTS

In this section, we provide additional qualitative results of 2D (Figure 20, 22) or 3D asset (Figure 21) creation based on PROOF. Figure 19 indicates the workable function of OTIB to conduct controllable diversity implicitly. Note that the detailed differences for small $\beta$ are not obvious. Please zoom in sufficiently and observe patiently.

**Model select**   As for PROOF_2D_Ref, we use Realistic_Vision_ V4.0_noVAE for diffusion inversion and denoising, with ip-adapter-plus_sd15 for appearance transfer. The VAE module is from stabilityai-stable-diffusion-2-1-base. In Figure 22, iRFDS+Instantx uses the checkpoint of InstantX-SD3.5-Large-IP-Adapter. In Figure 9, images of PROOF_2D are synthesized based on the checkpoint of Stable Diffusion v2-1_512-ema-pruned. In Figure 18, we use stabilityai-stable-diffusion-xl-base-1.0.

Note that because of the strong constraints from the image condition of TRELLIS Xiang et al. (2025), there is little diverse space for direct PROOF_3D_Img. Therefore, we first synthesize the image variants based on PROOF_2D and then conduct 3D modeling based on the trellis-image-large model. Text-based PROOF_3D uses the trellis-text-xlarge model, as shown in Figure 21.

**User Study**   We invite 100 domain experts to conduct the user study. First, we briefly explain the highly correlated asset creation task. We suggest that users carefully observe the original content and generated image variants obtained by 6 state-of-the-art methods and our proposed PROOF. Each observed algorithm has 20 samples. These observers need to select the better image variant set from 3 aspects: (a) overall quality, (b) overall fidelity considering structure and appearance, (c) controllable diversity subject to the fidelity. The interface of our user study is shown in Figure 23.

Table 5: GENEVAL Ghosh et al. (2023) scores of different models. Robust PROOF preserves the semantic content well and exhibits higher text-image correctness v.s. naive noise interpolation.

| Model | Overall | Single object | Two object | Counting | Colors | Position | Color attribution |
|---|---|---|---|---|---|---|---|
| CLIP retrieval | 0.35 | 0.89 | 0.22 | 0.37 | 0.62 | 0.03 | 0 |
| minDALL-E | 0.23 | 0.73 | 0.11 | 0.12 | 0.37 | 0.02 | 0.01 |
| Stable Diffusion v1.5 | 0.43 | 0.97 | 0.38 | 0.35 | 0.76 | 0.04 | 0.06 |
| Stable Diffusion v2.1 | 0.5 | 0.98 | 0.51 | 0.44 | 0.85 | 0.07 | 0.17 |
| Stable Diffusion XL | 0.55 | 0.98 | 0.74 | 0.39 | 0.85 | 0.15 | 0.23 |
| IF-XL | 0.61 | 0.97 | 0.74 | 0.66 | 0.81 | 0.13 | 0.35 |
| Naive $\lambda$=0.85 | 0.61 | 0.96 | 0.67 | 0.54 | 0.80 | 0.23 | 0.46 |
| PROOF $\beta$=0.1 | 0.70 | 0.98 | 0.80 | 0.65 | 0.91 | 0.32 | 0.55 |
| PROOF $\beta$=0.01 | 0.72 | 0.98 | 0.83 | 0.67 | 0.92 | 0.35 | 0.57 |

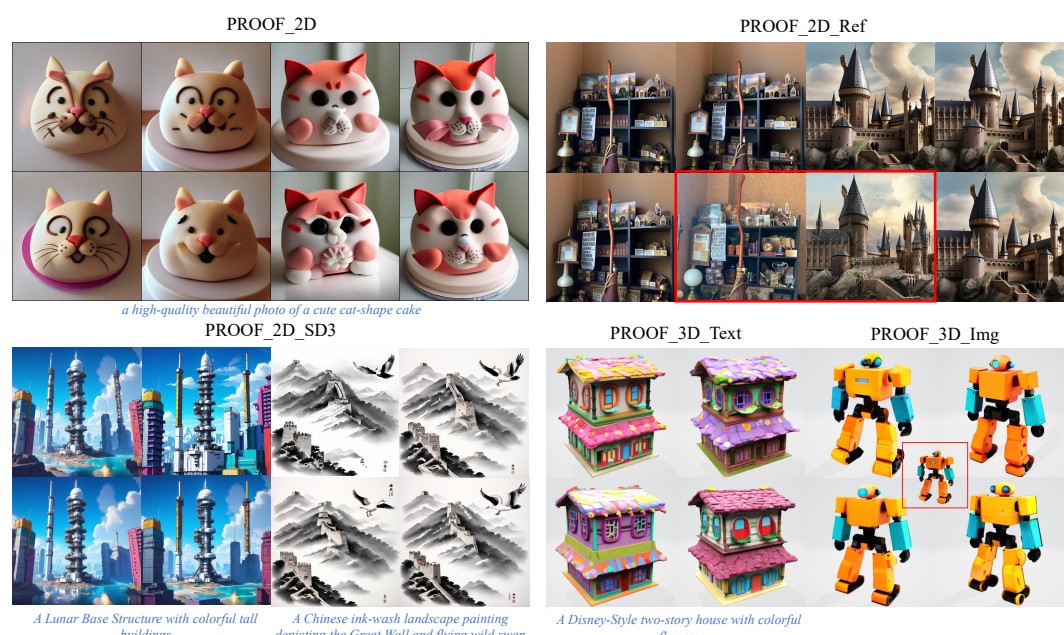

Figure 9: Our proposed PROOF is an effective learning framework to synthesize highly correlated assets where variants exhibit consistent structure and appearance. Test-time PROOF facilitates high-quality 2D assets Esser et al. (2024) and 3D assets Xiang et al. (2025) with high contextual fidelity and controllable diversity, under any text or image condition (red boxes).

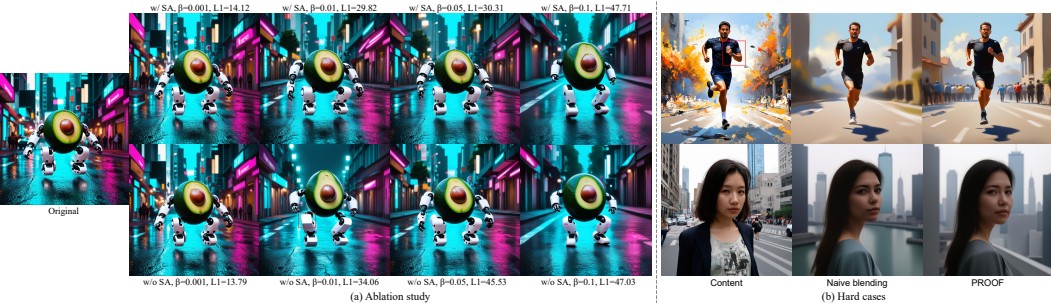

Figure 10: (a) PROOF variants show that methods w/ SA preserve better appearance statistics than those w/o SA. Higher $\beta$ usually intentionally relaxes contextual constraints but boosts the diversity. (b) The background lacks abundant details for large-scale information compression (e.g., $\lambda$=0.8), while the human identity and pose are maintained well.

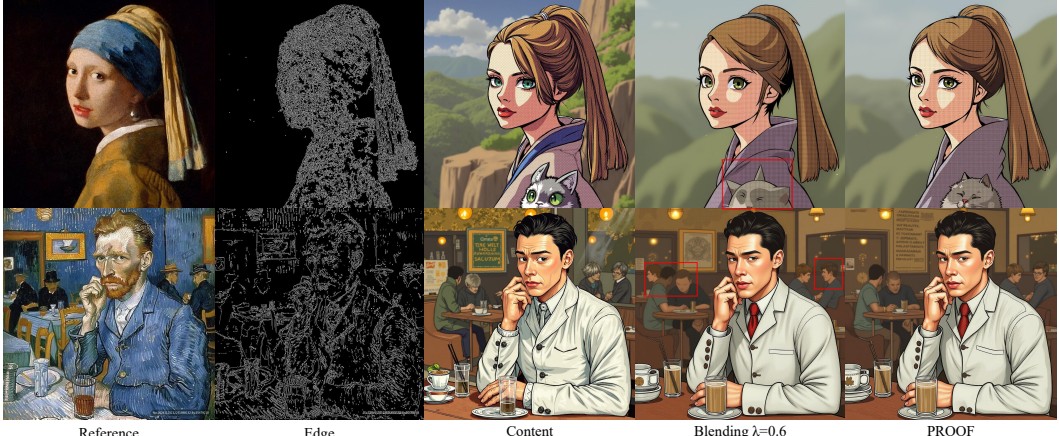

Figure 11: Integration of PROOF and a structure-guided controller. Despite being constrained by edge conditions, PROOF maintains structure and texture fidelity in local areas while still generating diverse variations. Under large-scale perturbation, PROOF performs robust variant generation.

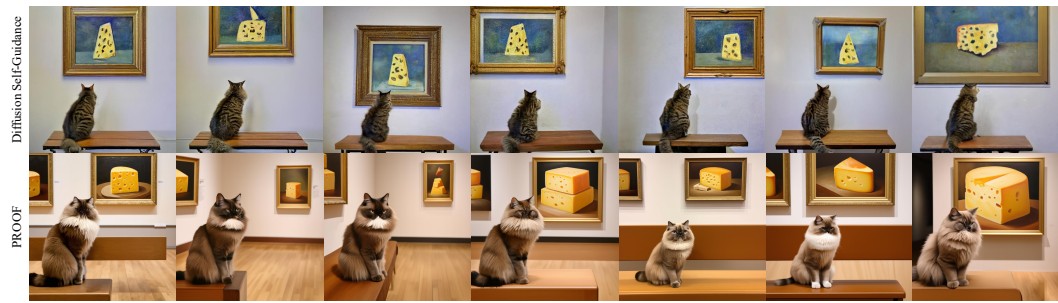

Figure 12: PROOF with semantic editing Mokady et al. (2023) produces high-quality editing results considering structure and appearance preservation.

**Comparision with DSG** While achieving similar editing effects to DSG Epstein et al. (2023) in Figure 13, our *PROOF* doesn't require any explicit guidance, e.g., position, size, shape.

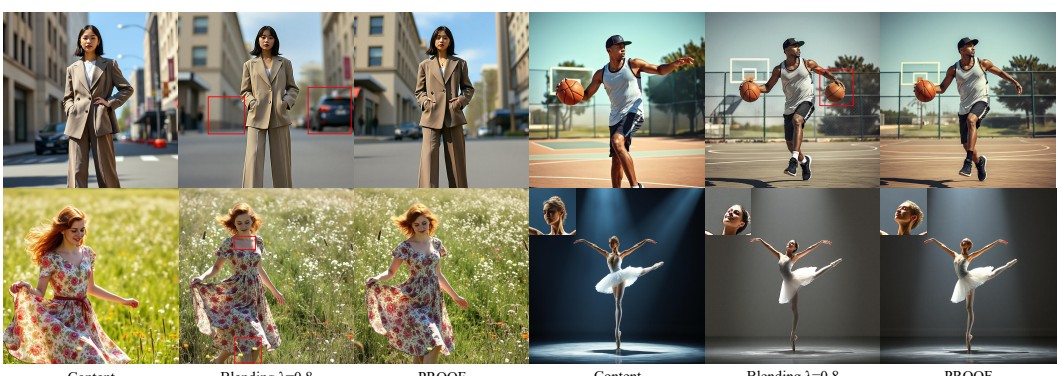

Figure 13: Feature workbench provided by DSG Epstein et al. (2023) is fine-grained but cumbersome. Our PROOF gives another efficient and diverse workbench to change the properties of objects.

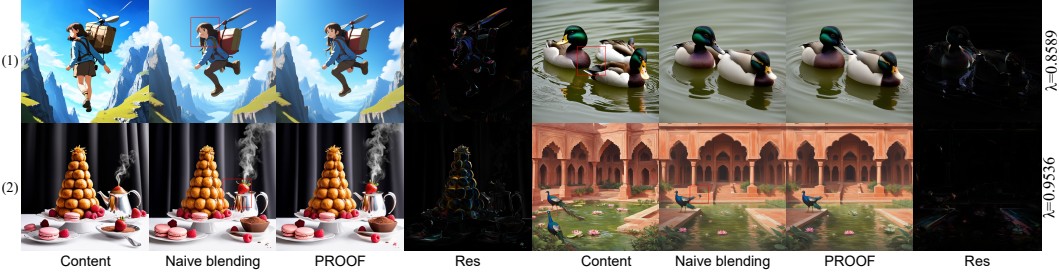

Figure 14: Content transformations based on FLUX.1-schnell with PROOF. We show some examples of human deformation with different poses and novel perspectives, which demonstrate that intrinsic interpolation to manipulate noise is efficient to model complex nonlinear transformation patterns. PROOF outperforms naive blending, as the latter often leads to noticeable content distortion and undesirable artifacts.

Figure 15: PROOF w/ $\beta = 0.1$ (Row 1) and $\beta = 0.05$ (Row 2) are corresponding with naive blending w/ $\lambda = 0.8589$ and $\lambda = 0.9536$, based on the mean value across the channel and spatial dimensions of PROOF's neural $\lambda$. PROOF preserves fine-grained structure and appearance features.

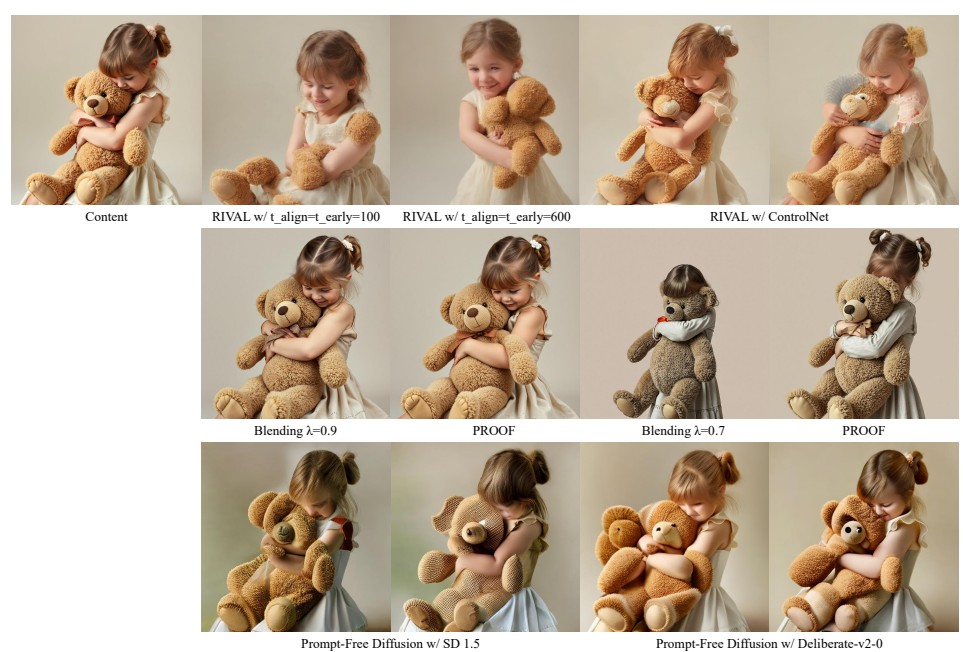

Figure 16: Variant comparison of PROOF and other image variation works. Zhang et al. (2023b) maintains alignment between the latent distributions of the generative and inverse paths to improve semantic and structural fidelity. Xu et al. (2024) eliminates the text encoder and text prompts, which may result in semantic degradation (e.g., face and teddy bear in Row 3). PROOF leveraging robust manifold manipulation preserves fine-grained structure and appearance features (Row 2). Moreover, adaptive interpolation via OTIB efficiently produces diverse high-fidelity image variants.

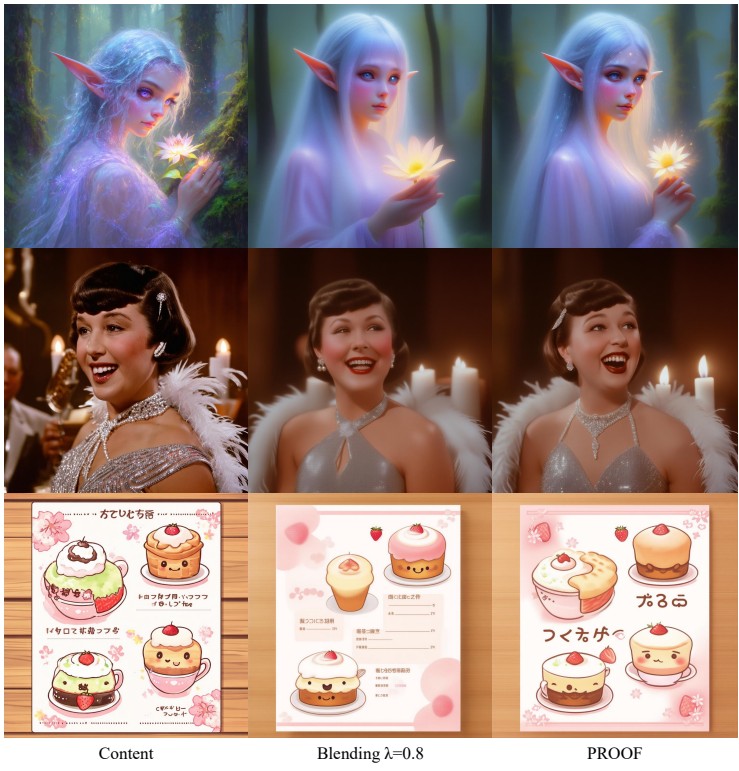

Figure 17: Additional visual results of PROOF_2D based on Stability AI SD3.5 Medium. PROOF is more robust to defend against noise perturbation.

**Comparison with Golden Noise** Task Differentiation of Golden Noise Zhou et al. (2025) and PROOF: Golden Noise focuses on text-embedding alignment in noise space and embeds semantic information into noise for semantic fidelity. PROOF targets content-aligned variation generation by modifying local structure and appearance distributions for contextual fidelity with diversity. We provide some comparative results in Fig. 18, which demonstrates that PROOF is powerful to synthesize high-fidelity and high-quality assets.

Specifically, given standard noise as $N_{Orig}$, we obtain golden noise $N_{Gold} = NPNet(N_{Orig}, c)$. Moreover, standard PROOF and golden PROOF are implemented based on $N_{Orig}$ and $N_{Gold}$, where the same $N_{Div}$ is adaptively interpolated via OTIB. Note that both NPNet and PROOF leverage SDXL as the pretrained base model.

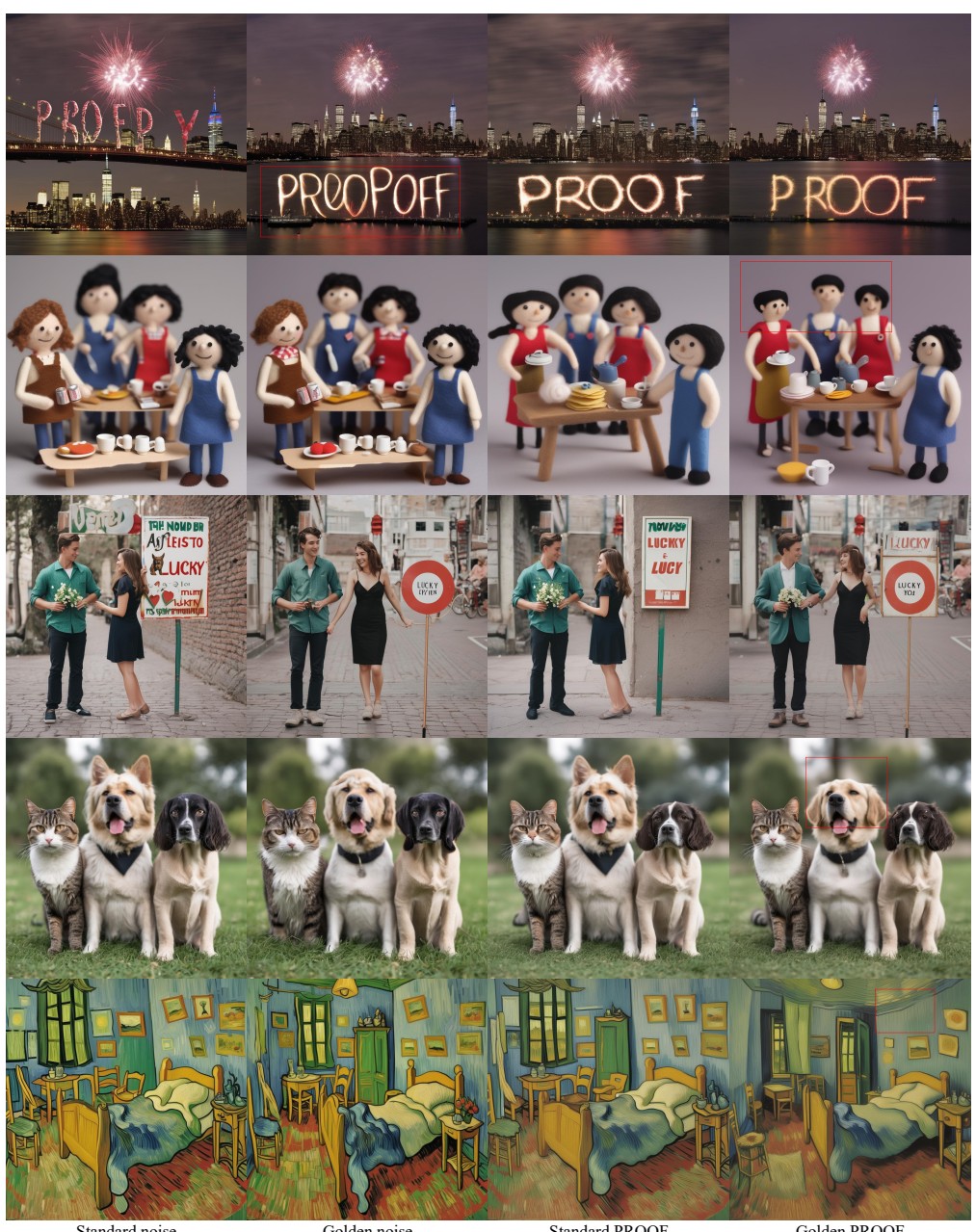

| Standard noise | Golden noise | Standard PROOF | Golden PROOF |

Figure 18: Standard PROOF and Golden PROOF are based on the standard noise and golden noise, respectively. PROOF seems to produce more high-fidelity golden noise (col 3), and Zhou et al. (2025) exhibits low perturbation robustness (col 4).

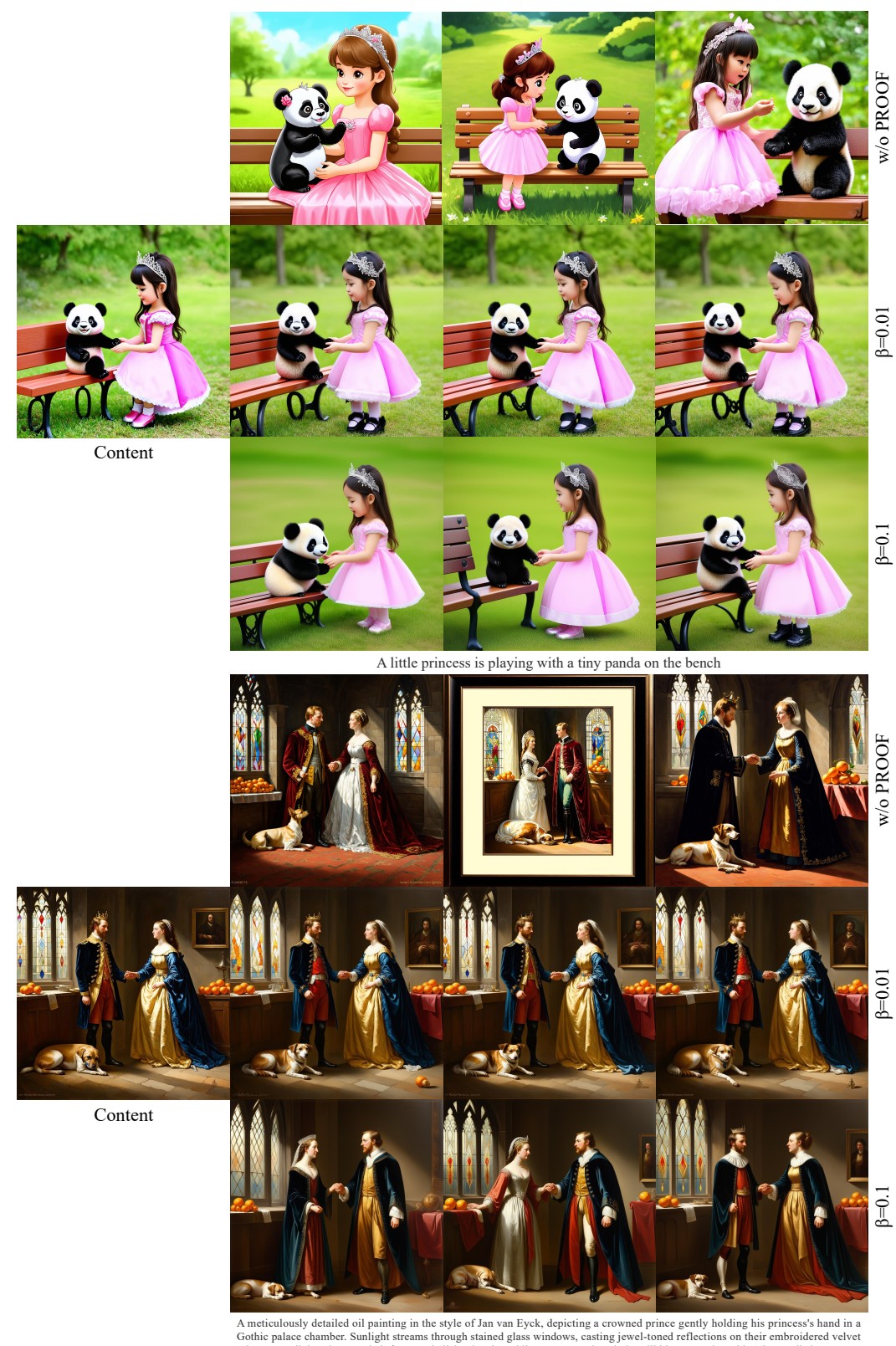

Figure 19: PROOF effectively controls the structure and appearance of the content. Smaller tradeoff weight $\beta$ puts content on a slight adjustment workbench, while larger $\beta$ changes the content more obviously, but maintains the scene layout.

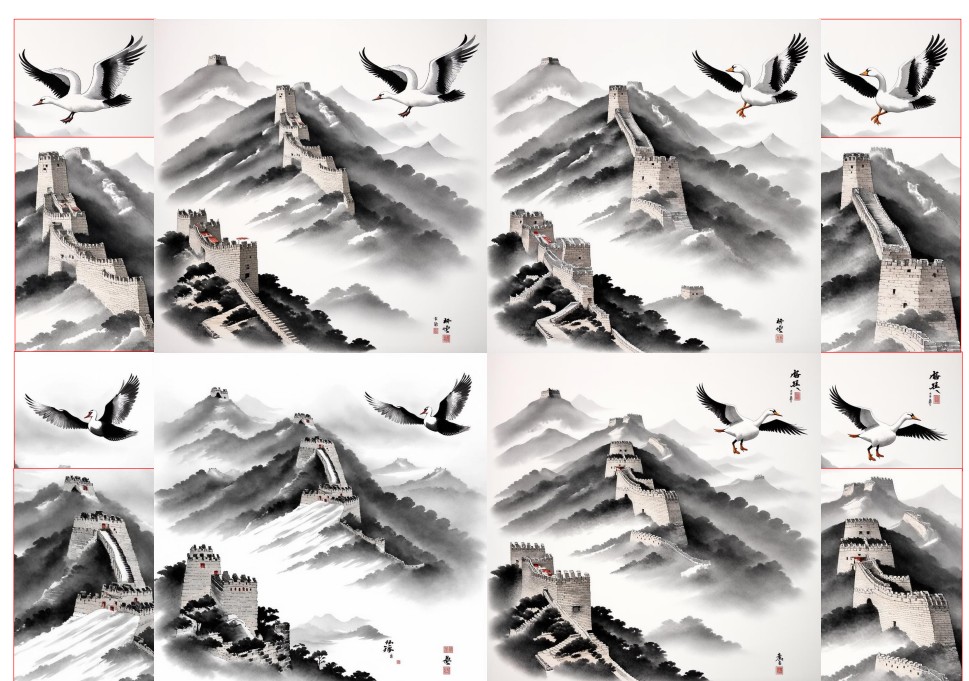

A Chinese ink-wash landscape painting depicting the Great Wall and flying wild swan, best quality

Figure 20: Image variants of the teaser figure 9 under magnified observation.

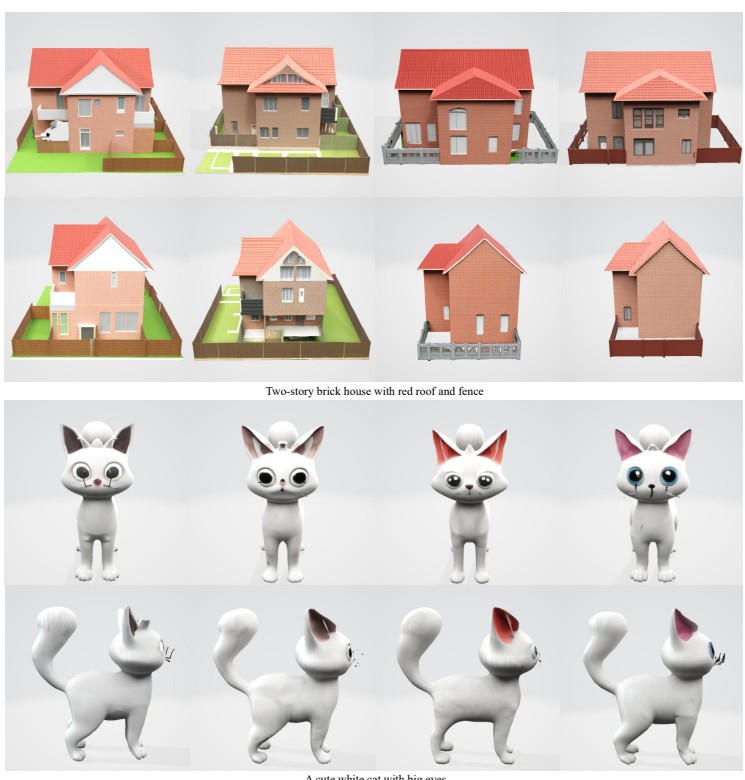

Two-story brick house with red roof and fence

A cute white cat with big eyes

Figure 21: More qualitative results of PROOF_3D based on TRELLIS Xiang et al. (2025).

1296
1297
1298

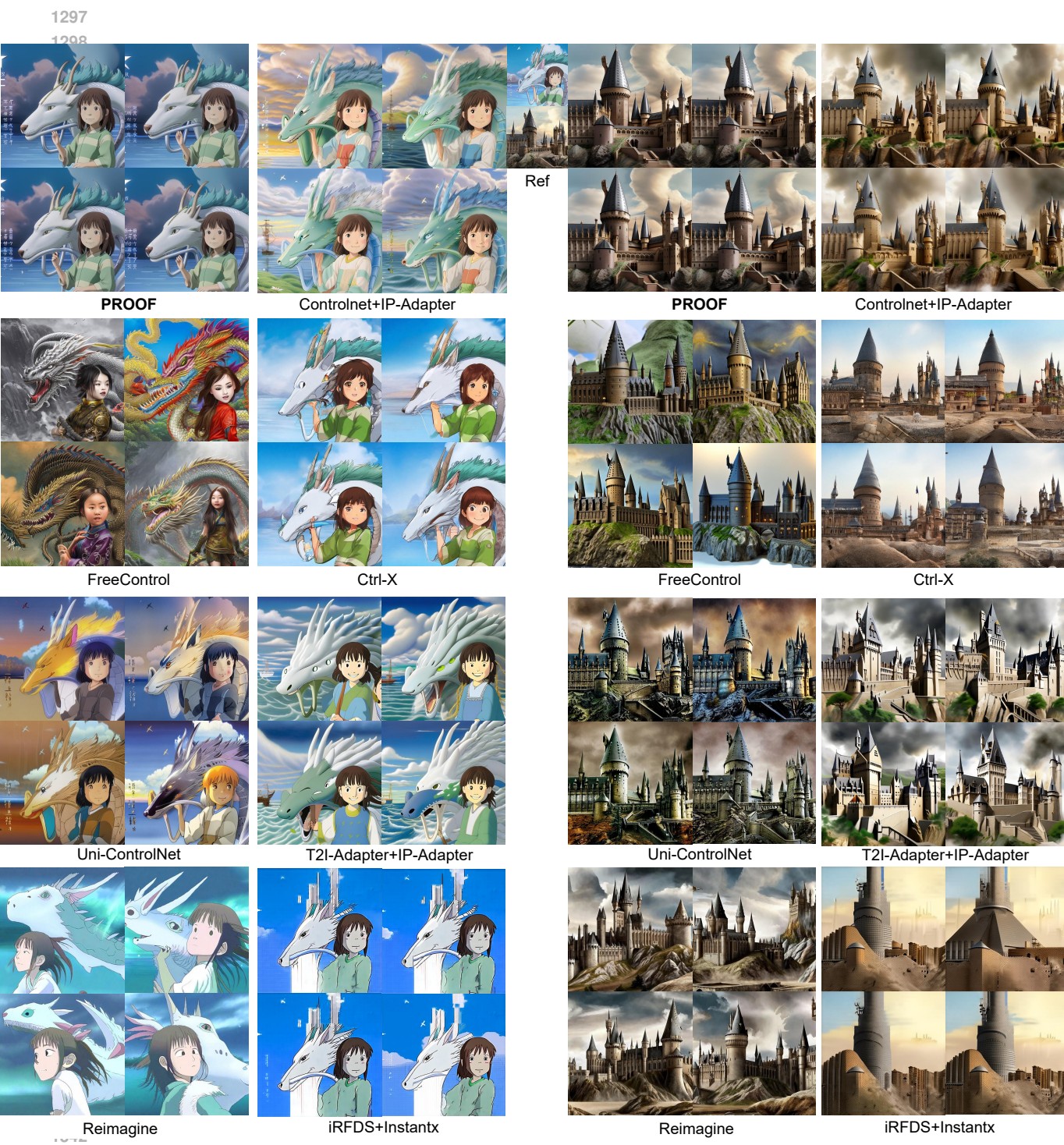

1343
1344
1345
1346
1347
1348
1349

Figure 22: Qualitative results of PROOF_2D_Ref, ControlNet Zhang et al. (2023a); Ye et al. (2023), FreeControl Mo et al. (2024), Ctrl-X Lin et al. (2024), Uni-ControlNet Zhao et al. (2023), T2I-Adapter Mou et al. (2024); Ye et al. (2023), Reimagine AI (2023) and iRFDS Yang et al. (2025) on the wild images.

Figure 23: (a) Additional qualitative results of PROOF_2D_Ref, ControlNet Zhang et al. (2023a); Ye et al. (2023), FreeControl Mo et al. (2024), Ctrl-X Lin et al. (2024), Uni-ControlNet Zhao et al. (2023), T2I-Adapter Mou et al. (2024); Ye et al. (2023), and Reimagine AI (2023). (b) The interface of our user study.

