# OpenReview forum: "PROOF: Perturbation-Robust Noise Finetune via Optimal Transport Information Bottleneck for Highly-Correlated Asset Generation"
_ICLR.cc/2026/Conference — Submitted to ICLR 2026_

### Official Review · Reviewer_b8eS · 2025-10-27

**Soundness:** 3
**Presentation:** 2
**Contribution:** 2
**Rating:** 4
**Confidence:** 4

**Summary:**

This paper introduces PROOF, a lightweight plug-and-play module that improves the balance between fidelity and diversity in diffusion-based generation by directly fine-tuning latent noise. The method uses a new Optimal-Transported Information Bottleneck (OTIB) to adaptively blend original and perturbed noise, preserving topology and texture while enabling controllable variation. PROOF requires minimal training and works across 2D/3D diffusion models. Experiments show better content consistency and diversity compared to existing controllable generation approaches.

**Strengths:**

This paper introduces a novel and practical perspective by leveraging noise itself as a controllable representation for diffusion models, rather than relying on explicit structural or textural signals. The method is theoretically supported through a closed-form Sinkhorn-IB solution and is lightweight and model-agnostic, enabling efficient plug-and-play usage across 2D/3D generators.

**Weaknesses:**

1. [1] demonstrates not only image variation but also image editing without introducing any additional modules or training, whereas PROOF still requires lightweight training and does not explicitly support localized editing with semantic prompts. This raises the question of whether the proposed method offers enough practical advantage over fully training-free approaches.

2. The comparisons are mainly focused on diffusion–control based methods, and there is still room for stronger empirical validation through comparison with image variation works such as [1] and [2], which also aim at structure- and content-preserving diversity.

3. In the main text, the explanation of the Experimental Results appears only on page 9. Apart from the tables and figures, there is insufficient discussion of the experimental findings, which made it difficult to fully understand the paper. I believe the paper should provide a better presentation. For example, in the Section "5 Experiments", the descriptions of the Training Protocol and Baselines could, in my view, be largely moved to the Appendix.

[1] Real-World Image Variation by Aligning Diffusion Inversion Chain
[2] Prompt-Free Diffusion: Taking “Text” out of Text-to-Image Diffusion Models

**Questions:**

1. Can the proposed method perform image editing by conditioning on a different prompt, or is the diversity strictly limited to preserving the original semantic content?

2. Does the method works on latest flow models like SD3.5/Flux?

3. When combining PROOF with a structure-guided controller such as ControlNet, is it possible to maintain texture fidelity while still generating diverse variations?

4. Why is the inference time in Table 1 presented in two different types only for “Ours”? I could not find a clear explanation for this in either the main text or the caption.

---

> ### Author Response · Authors · 2025-11-20
>
> **Response 1**: We sincerely appreciate the reviewer's insightful feedback. The lightweight training based on the closed-form OTIB solution is required for high-fidelity and highly-correlated asset generation. Furthermore, Fig. 12 on page 20 of the revised paper shows an example of localized semantic editing with PROOF.
>
> Fig. 16 provides the variant comparison of PROOF and RIVAL. PROOF leveraging robust manifold manipulation preserves fine-grained structure and appearance features. Moreover, adaptive interpolation via OTIB efficiently produces diverse high-fidelity image variants.
>
> **Response 2**: We sincerely appreciate the reviewer's critical perspective on our method comparison. We have included the quantitative and qualitative experiments in Table 1 and Fig. 16, compared with RIVAL and prompt-free diffusion. PROOF leveraging robust manifold manipulation preserves more fine-grained structure and appearance features.
>
> **Response 3**: Thanks for your valuable suggestions. We have added more analyses in the Experiments section of the revised paper, including content transformation, train-test resolution discrepancy, local variation, DiT-based model generalization, along with tradeoff weight $\beta$. And the descriptions of the Training Protocol and Baselines have been largely moved to Appendix B and C.
>
> **Response 4**: PROOF study the robust noise representation learning for highly-correlated asset generation. The high correlation means the semantic information among diverse variants is shared and similar. If the semantic content is not strictly preserved, we think this scenario is weakly-correlated because of the structure and appearance discrepancy.
>
> Users could use specific semantic editing methods, e.g., prompt-to-prompt [1], to perform the semantic editing, and then utilize PROOF for high-quality content preservation and diversity.
>
> [1] Hertz A, Mokady R, Tenenbaum J, et al. Prompt-to-prompt image editing with cross attention control[J]. arXiv preprint arXiv:2208.01626, 2022.
>
> **Response 5**: We have provided plenty of experiments based on SD3.5 (Fig. 17) and Flux (Figures 4, 5, 7, 11, 14) in the revised paper.
>
> **Response 6**: It's applicable to conduct this kind of controllable variation generation based on PROOF. We give an example in Fig. 11 on page 19 of the revised paper. Despite being constrained by edge conditions, PROOF maintains structure and texture fidelity in local areas while still generating diverse variations.
>
> **Response 7**: We sincerely appreciate the reviewer's insightful feedback. The previous inference time in Table 1 contains PROOF_2D and PROOF_2D_Ref, i.e., 7.3 / 27.2. The latent noise is available for PROOF_2D, while the latent inversion of PROOF_2D_Ref is time-consuming. In the revised paper, we delete the PROOF_2D time for more clear presentation.

---

### Official Review · Reviewer_UeUB · 2025-10-30

**Soundness:** 1
**Presentation:** 2
**Contribution:** 2
**Rating:** 4
**Confidence:** 3

**Summary:**

The paper introduces PROOF, a plug-and-play module for diffusion models based on an Optimal Transport Information Bottleneck formulation. The method fine-tunes latent noise through an optimal transport constraint to balance content fidelity and diversity, with claimed benefits in robustness and controllability.

While mathematically sound, the method’s practical contribution is limited. The approach introduces additional computational complexity without offering clear performance advantages, which undermines its claimed efficiency. It remains unclear why practitioners should adopt this method.

**Strengths:**

* The integration of optimal transport and information bottleneck concepts is theoretically consistent.
* The framework is modular and, in principle, applicable to different diffusion architectures.

**Weaknesses:**

* **Inferior inference efficiency.** The paper claims PROOF is a lightweight, plug-and-play controller. However, Table 1 shows that its inference time is longer than all compared baselines. This demonstrates that the method adds computational overhead rather than reducing it, which makes it impractical for real-time or large-scale applications.

* **Modest empirical improvements.** The reported results show only minor gains over existing baselines, which do not justify the added architectural complexity or computational cost.

* **Sensitivity concerns.** The method relies on several manually tuned hyperparameters, raising questions about its robustness, reproducibility, and generalization across different datasets and architectures.

**Questions:**

* Why should practitioners adopt PROOF over simpler, faster alternatives that yield similar results?
* Can the method be adapted to few-step models (e.g., FLUX-Schnell) to reduce latency?
* Can the method be evaluated on modern diffusion models (e.g., SD3.5, FLUX) to ensure its relevance?

---

> ### Author Response · Authors · 2025-11-20
>
> **Response 1**: We sincerely appreciate the reviewer's insightful feedback. Comprehensive experiments have demonstrated that PROOF is a lightweight, plug-and-play controller for high-fidelity asset creation. Note that 27.2s in Table 1 means the total inference time of ROOF_2D_Ref, where diffusion inversion is time-consuming, nearly 24s for Null-Text Inversion [1] with 50 DDIM steps.
>
> As for PROOF_2D, the computational cost mainly comes from the SD or Flux. For example, the inference time of SD-3 with 28 inference steps on NVIDIA 4090 takes around 7s. While the inference of noise finetuning based on OTIB exhibits low computational cost and small trainable parameters as follows.
>
> | Models   | Spatial latent      | MACs       |Params  | Inference time |
> |--------|------------|---------|---------|--------------|
> | SD-1.4, SD-1.5, SD-2 |(B, 4, 64, 64) | 134.64 MMac | 100     | 0.1579s      |
> | SD-3, SD-3.5, Flux | (B, 16, 128, 128)| 8.61 GMac  | 1.36 k  | 0.2185s      |
>
> [1] Mokady R, Hertz A, Aberman K, et al. Null-text inversion for editing real images using guided diffusion models[C]//Proceedings of the IEEE/CVF conference on CVPR. 2023: 6038-6047.
>
> **Response 2**: We sincerely appreciate the reviewer's insightful feedback. Particularly in strong perturbation scenarios with low $\lambda$, PROOF improves local content fidelity obviously, for example, the structure and appearance preservation, as shown in Figs 4, 5, 7, 11, 16, 17, 18 of the revised paper.
>
> **Response 3**: We sincerely appreciate the reviewer's insightful feedback. As shown in Fig. 7 of the revised paper, there is basically a complementary relationship between the trade-off hyperparameter β and the average interpolation weight λ. Moreover, comprehensive experiments demonstrate PROOF's robustness and generalization across different datasets and architectures.
>
> **Response 4**: We sincerely appreciate the reviewer's insightful comment regarding the performance of naive alternative and PROOF. As shown in Figs 4, 5, 6, 14, 16, there are obvious, unreasonable artifacts on the generative results of naive noise blending. To synthesize high-fidelity and diverse variants, PROOF adopts OTIB for a better trade-off between content preservation and diversity.
>
> **Response 5**: We sincerely appreciate the reviewer's insightful feedback. Figures 4 and 5 are based on FLUX-Schnell. Remarkably, as mentioned in Response 1, although the large model itself is the primary source of latency, OTIB delivers acceptable inference performance.
>
> **Response 6**: We sincerely appreciate the reviewer's insightful feedback. We have included the corresponding experiments based on Flux (Fig. 4, 5, 7, 11, 14) and SD3.5 (Fig. 17). It works for these DiT-based diffusion models.

---

> > ### Comment · Reviewer_UeUB · 2025-11-27
> > **Official Comment by Reviewer UeUB**
> >
> > Thanks for the reply.
> >
> > While the authors address my concern regarding broader applicability, **I still do not understand why this approach would be useful in practice and what its concrete applications are.** The authors show improved blending results, but the purpose of this blending and why it is needed remains unclear.
> >
> > As the authors suggest in
> > line 537: *“Our method provides a robust editor for both images and 3D models”*,
> > they list several potential applications, mainly related to image editing (novel view rendering, Figure 14; local editing, Figure 5; semantic editing, Figure 12). However, these examples are not convincing, and no numerical results are provided.
> >
> > **If the authors want to demonstrate that their approach is genuinely useful, they should compare it against state-of-the-art methods or apply their approach on top of them to demonstrate the usefulness of the proposed blending approach in real cases: [1], [2], [3], [4], [5], [6].**
> >
> > The only baseline they consider is [7], which is significantly outdated.
> >
> > [1] FLUX.1 Kontext: Flow Matching for In-Context Image Generation and Editing in Latent Space
> >
> > [2] Qwen-Image Technical Report
> >
> > [3] Omnicontrol: Minimal and Universal Control for Diffusion Transformers
> >
> > [4] Step1x-Edit: A Practical Framework for General Image Editing
> >
> > [5] Emerging Properties in Unified Multimodal Pretraining
> >
> > [6] TurboEdit: Text-Based Image Editing Using Few-Step Diffusion Models
> >
> > [7] Null-text Inversion for Editing Real Images Using Guided Diffusion Models
> >
> > There are many leading approaches that can already accomplish these tasks without any fine-tuning. **Therefore, I keep my score unchanged and maintain a negative overall assessment of the paper.**

---

> > > ### Author Response · Authors · 2025-11-28
> > >
> > > Thanks for the reply.
> > >
> > > ### 1. Application Scenario
> > > For creative tasks with fixed semantics, given several noise seeds, users may prefer one generated asset. When additional diversity with overall content preservation is required, **PROOF serves as a powerful fine-grained editing tool to synthesize highly-correlated variants with structure and appearance robustness under strong perturbations**.
> > >
> > > ### 2. Unfairness in Comparisons
> > > **[1,2,4,5] train their model based on millions of high-quality data, so comparing them with our paper (which only performs lightweight training in the noise latent space without other diffusion modifications) is unfair**. Moreover, [1-6] do not investigate the robustness of noise fine-tuning under perturbations - **they have significant differences in research focus with PROOF**.
> > >
> > > - FLUX.1 Kontext[1] modifies the diffusion mechanism through latent token concatenation and virtual time steps.
> > > - Qwen-Image[2], Step1X-Edit[4], and BAGEL[5] use carefully collected data for large-scale parameter training, enabling powerful image editing capabilities.
> > > - OminiControl[3] modifies the attention operation of Diffusion Transformer.
> > > - TurboEdit[6] focuses on the text-based semantic editing task rather than highly-correlated variant generation.
> > >
> > > ### 3. PROOF Effectiveness and Practicality
> > > PROOF has natural generalization in both theory and practice. As a noise manipulation method, it can plug-and-play into various image generation and editing models (SD, FLUX), which have been extensively validated in our paper. **PROOF is also equally applicable for the mentioned [1-6], as these advanced models are still based on SD or DiT architectures**.
> > >
> > > ### 4. Broader Impact
> > > This work focuses editing scope on Gaussian noise, **utilizing optimal transport information bottleneck for robust noise editing**. This represents a novel and insightful research contribution.

---

### Official Review · Reviewer_B9Aq · 2025-10-31

**Soundness:** 3
**Presentation:** 3
**Contribution:** 3
**Rating:** 6
**Confidence:** 2

**Summary:**

This paper proposes PROOF, a controllable 2D/3D generation method based on perturbation-robust noise finetuning. By integrating information bottleneck with optimal transport theory, the paper derive a closed-form solution for Sinkhorn-regularized interpolation weights. It achieves high-quality topology and texture-aligned generation across multiple base models without requiring external control signals, while outperforming existing methods on both fidelity and diversity metrics.

**Strengths:**

1. The paper's significant strenghts  is its theoretical grounding, as the closed-form solution derived via Optimal-Transport Information Bottleneck (OTIB) provides clear guidance for subsequent optimization.

2.  In terms of general applicability, the method is compatible with both 2D and 3D generation tasks, supports text and image guidance modalities, and can be integrated with multiple base models (including Stable Diffusion and TRELLIS).

3. Compared to existing control methods, PROOF operates without relying on any external structural control signals and achieves superior appearance alignment performance without requiring personalized concept data or model fine-tuning.

**Weaknesses:**

1. Although diversity is controlled through the β parameter, its variation range remains constrained by the structure of the noise space. Could we further explore the implications of β parameters for generation diversity?

2. I'm concerned about relying on intrinsic interpolation to manipulate noise, as it fails to capture complex nonlinear content transformations—such as object deformation or perspective changes. This interpolation paradigm constitutes a fundamental limitation in scenarios demanding highly creative generation.

**Questions:**

1. Its potential applicability to more advanced diffusion models with distinct architectures—such as Flux or SD3.5—warrants further investigation. Conducting experiments on such next-generation models would significantly strengthen PROOF's generalizability and broader relevance.

2. Does PROOF maintain robust performance when there is a resolution discrepancy between the fine-tuning and inference phases? For instance, when fine-tuning employs low-resolution data while inference utilizes high-resolution data.

3. Does Sinkhorn suffer from over-smoothing phenomena? For instance, if I intend to perform texture stylization only on specific local regions rather than globally – in such tasks, critical information may be confined to local areas. When processing a human face, the eye regions should inherently receive greater attention, yet Sinkhorn enforces uniform distribution. Could this potentially weaken the strong correlations between adjacent pixels in texture synthesis?

---

> ### Author Response · Authors · 2025-11-20
>
> **Response 1**: We sincerely appreciate the reviewer’s constructive feedback. The variation range actually is large, where $\lambda$ is from 0.99 to 0.5 or less. Moreover, different noise perturbations could be interpolated for diverse variants. We have explored the implications of β parameters for generation diversity in Fig. 7 on page 10 of the revised paper. PROOF exhibits obvious diversity over the parameter range.
>
> **Response 2**: We sincerely appreciate the reviewer's insightful feedback regarding nonlinear content transformations. Although PROOF applies intrinsic interpolation to manipulate noise, the latent space compressed by VAE is already a high-dimensional manifold where nonlinear content transformations are represented to some extent. That means the change of a specific noise point with a certain channel and position is capable of imposing contextual transformation on several image-level areas, therefore visually leading to object deformation or novel-view rendering. The visual examples are shown in Fig. 14 on page 20 of the revised paper.
>
> **Response 3**: We sincerely appreciate the reviewer's insightful feedback. We have included the corresponding experiments based on Flux (Fig. 4, 5, 7, 11, 14) and SD3.5 (Fig. 17). It works for these DiT-based diffusion models.
>
> **Response 4**: We thank the reviewer for raising these important points about train-test resolution discrepancy. The overall image contents based on different finetuning models are consistent. However, the finetuning model employing 32/64-resolution data hardly captures local topological and textural details when dealing with 128-resolution data inference. The qualitative results are illustrated in Fig. 4 on page 8 of the revised paper.
>
> **Response 5**: We appreciate the reviewer's insightful observations about the uniform distribution of Sinkhorn. It is worth noting that due to the high-degree feature entanglement in the high-dimensional manifold space, implementing a uniform attention distribution proves effective. Global and comprehensive attention to the Gaussian noise latent is encouraged and will result in better content fidelity. Some local editing examples are in Fig. 5 on page 9 of the revised paper.

---

### Author Response · Authors · 2025-12-01

We thank all reviewers for their positive assessment and valuable feedback. We are pleased that the reviewers recognize:

**Key Strengths Acknowledged by Reviewers:**
- **Theoretical grounding** with closed-form OTIB solution providing clear optimization guidance (R1)
- **General applicability** across 2D/3D tasks, multiple guidance modalities, and base models including Stable Diffusion and TRELLIS (R1)
- **Theoretical consistency** in integrating optimal transport with information bottleneck (R2)
- **Novel perspective** of using noise as a controllable representation instead of explicit signals (R3)
- **Practical advantages** of being lightweight, model-agnostic, and plug-and-play (R3)

**Addressing Reviewers' Concerns:**
In response to all raised concerns, we have:

1. **Provided additional experiments** validating the algorithm's generalizability and broader relevance, including compatibility with Flux and SD3.5. Moreover,  PROOF is also equally applicable to the methods mentioned by R2, as these advanced models are still based on SD or DiT architectures.
2. **Demonstrated theoretical rationality** of OTIB through comprehensive analysis
3. **Shown practical combination** with various controllers while maintaining robustness
4. **Verified robust content preservation** under strong perturbations
5. **Established superiority** over existing image variation works

**Conclusion:**
This work focuses editing scope on Gaussian noise, utilizing optimal transport information bottleneck for robust noise editing. The method's theoretical foundation, practical effectiveness, and broad applicability represent a novel and insightful research contribution to the field.

---

### Meta-Review · Area_Chair_BM3o · 2026-01-05

**Summary:**

This work proposes an interesting perspective, treating latent Gaussian noise as a controllable representation, so that this work has a theoretical formulation that combines optimal transport and information bottleneck and includes a closed-form OTIB/Sinkhorn-IB solution that guides the optimization. Reviewers acknowledge broad applicability claims across 2D/3D settings, multiple guidance modalities, and multiple base diffusion/DiT models, with the revised manuscript adding experiments on FLUX and SD3.5.

However, the reviews revealed a key split: while one reviewer is marginally positive and another sees the contribution as potentially useful, the most critical reviewer argues that the work’s practical value and application justification remain unclear, especially given: (i) added complexity and latency, (ii) only modest empirical improvements, and (iii) a lack of comparisons to strong, modern editing/variation baselines, making it difficult to assess whether the method is compelling beyond the paper’s chosen setting (highly-correlated variant generation under strong noise perturbations). The rebuttal and revision address several technical questions (e.g., experiments on Flux/SD3.5 and clarification of timing), but the most critical reviewer remains unconvinced about usefulness and competitiveness. Overall, the paper provides a promising idea but still falls short on substantiating compelling impact and adoption-level motivation. Thus, a rejection is recommended.

**Reviewer Concerns:**

Concerns addressed by rebuttal
* Generalizability to modern models (Flux, SD3.5): The authors added experiments on Flux and SD3.5 in the revision, addressing repeated reviewer requests for modern-architecture validation.
* Clarification of inference-time reporting: The authors clarified that the large runtime number corresponds to a reference pipeline involving diffusion inversion (e.g., Null-Text Inversion), and that OTIB finetuning itself is relatively small compared to base model inference.
* β / diversity trade-off: The authors expanded analysis of β and show diversity variation across a wider range, addressing the request for clearer diversity control evidence.
* Train/test resolution discrepancy: The revision includes qualitative evidence and explanation of behavior under resolution mismatch (global structure preserved, but limited recovery of fine details when trained at low resolution).
* Presentation / experimental discussion: The authors claim improved experiment-section analysis and moved protocol/baseline details to appendix, which directly addresses a presentation weakness raised by one reviewer.

Concerns still outstanding
* Unclear practical motivation and application value: The strongest negative feedback remains that it is still unclear why practitioners would adopt PROOF over simpler or training-free alternatives. The paper shows improved “blending/variation” behaviors, but the motivation and concrete value proposition remain insufficiently convincing.
* Insufficient comparisons to strong, modern editing/variation baselines: Multiple reviewers requested comparisons to stronger and more recent methods. The authors argue such comparisons are “unfair” due to training/data scale differences and focus mismatch. While partially reasonable, this does not eliminate the need for stronger empirical positioning: either (i) competitive baselines in the same task setting, (ii) integration on top of SOTA methods, or (iii) clearer evidence that PROOF uniquely solves a well-motivated niche problem.
* Modest improvements vs. added overhead/complexity: Even with clarification that OTIB itself is not the dominant cost, reviewers remain unconvinced that the gains justify the method’s complexity. Improvements are presented as incremental in places, and the paper does not yet establish a compelling “must-have” scenario.

**Reviewer Scores:**

Reviewer B9Aq (6 → 6): addresed by the experiments and discussion.

Reviewer UeUB (4 → 4): This reviewer explicitly followed up after rebuttal and stated they keep their score unchanged, maintaining a negative assessment focused on unclear utility and insufficient SOTA comparisons.

Reviewer b8eS (4 → 4 or a bit higher): Many concrete concerns were addressed (Flux/SD3.5 tests, inference-time clarification, improved experiment discussion, added comparisons to variation baselines). This reviewer would likely increase slightly, though it is unclear they would cross into clear accept without stronger SOTA comparisons.

---

### Decision · Program_Chairs · 2026-01-26

Reject